# ASPEN, a methodology for reconstructing protein evolution with improved accuracy using ensemble models

Roman Sloutsky[1,2,3,4], Kristen M Naegle[2,4,5,6]*

[1]Program in Computational and Systems Biology, Washington University, St. Louis, United States; [2]Department for Biomedical Engineering, Washington University, St. Louis, United States; [3]Department of Biochemistry and Molecular Biology, University of Massachusetts, Amherst, United States; [4]Center for Biological Systems Engineering, Washington University, St. Louis, United States; [5]Department of Biomedical Engineering, University of Virginia, Charlottesville, United States; [6]Center for Public Health Genomics, University of Virginia, Charlottesville, United States

**Abstract** Evolutionary reconstruction algorithms produce models of the evolutionary history of proteins or species. Such algorithms are highly sensitive to their inputs: the sequences used and their alignments. Here, we asked whether the variance introduced by selecting different input sequences could be used to better identify accurate evolutionary models. We subsampled from available ortholog sequences and measured the distribution of observed relationships between paralogs produced across hundreds of models inferred from the subsamples. We observed two important phenomena. First, the reproducibility of an all-sequence, single-alignment reconstruction, measured by comparing topologies inferred from 90% subsamples, directly correlates with the accuracy of that single-alignment reconstruction, producing a measurable value for something that has been traditionally unknowable. Second, topologies that are most consistent with the observations made in the ensemble are more accurate and we present a meta algorithm that exploits this property to improve model accuracy.
DOI: https://doi.org/10.7554/eLife.47676.001

*For correspondence:
kmn4mj@virginia.edu

## Introduction

Orthology and paralogy are two forms of evolutionary homology between genetic sequences introduced by Walter Fitch (*Fitch, 2000*; *Koonin, 2005*) to distinguish between two modes of descent from a common ancestor. Homologs diverged through speciation are called orthologs, while those diverged through the duplication of a genomic region are called paralogs. For protein coding sequences, paralogs tend to perform related, but distinct functions (*Prince and Pickett, 2002*; *Raes and Van de Peer, 2003*; *Espinosa-Cantú et al., 2015*). As a result, families of paralogous proteins provide an excellent opportunity for biochemists and molecular biologists to dissect how functionality is encoded in sequence and structure. Reconstructing histories of paralog divergence can aid tremendously in this endeavor (*Thomson et al., 2005*; *Eick et al., 2012*; *Rahman et al., 2014*; *Wilson et al., 2015*; *Daugherty et al., 2016*; *Harms and Thornton, 2013*). However, such reconstructions can be difficult due to a variety of factors, including, among others: the computational complexity of the problem, which scales combinatorially with the number of nodes to be reconstructed; the failure of likelihood-based approaches to adequately discriminate between topology models, especially when the amount of phylogenetic data (number of alignment columns) is small (<1000 columns) (*Yang and Zhu, 2018*); and the sensitivity of phylogenetic reconstruction to the

input alignment (*Ogden and Rosenberg, 2006*; *Landan and Graur, 2007*; *Wong et al., 2008*; *Liu et al., 2010*; *Wang et al., 2011*; *Blackburne et al., 2013*; *Sievers et al., 2013*). The current state-of-the-art in addressing the effect of alignment uncertainty on phylogenetic reconstruction, joint Bayesian inference of alignment and phylogeny with BAli-Phy (*Redelings and Suchard, 2005*; *Suchard and Redelings, 2006*), is far too computationally expensive to tackle large-scale reconstruction of paralog divergence. In this work, we propose a new framework for addressing the challenges in reconstruction of protein family divergence. Our approach takes advantage of model sensitivity to inputs to improve the accuracy of reconstruction.

We propose to reduce the complexity of reconstruction, motivated by the fact that not all ancestral nodes in protein evolution are of equal interest. Since a typical collection of input sequences contains both orthologs and paralogs, the divergence topology for these sequences contains both duplication and speciation ancestral nodes. Depending on the protein family and the number of species represented in the collection, the fraction of speciation nodes may be quite high. Reconstructing speciation history from a single protein family is of little interest aside from identifying extremely rare evolutionary events, such as horizontal gene transfer and reciprocal paralog loss, through reconciliation of the protein phylogeny with the accepted species phylogeny (*Nakhleh, 2013*; *Kamneva and Ward, 2014*). And yet, all speciation nodes must be inferred in order to reconstruct any duplication nodes that predate them. We propose sacrificing reconstruction of ancestral nodes of little interest to make reconstruction of ancestral nodes of greatest interest more robust.

Given any high confidence ancestral node, we can imagine "factoring'' the search of topology space into two components: (a) topologies below the node in question, (b) everything else. This is not particularly helpful for phylogenetic reconstruction in general, since (a) and (b) still must be optimized jointly in order to determine the overall optimal phylogeny. However, if we were not concerned with inferring (a), we would be satisfied to integrate over the uncertainty of (a) to produce, in a sense, a marginal reconstruction of (b). Moreover, treating the root of (a) – the high confidence ancestral node – as a leaf in our reconstruction of (b), would allow us to simultaneously consider multiple sources of uncertainty in the "integrated out'' reconstruction (a): the selection of specific sequences descended from that ancestor to be included in the reconstruction, the alignment of those sequences, and the uncertainty in the inference of (a), given that alignment. Here, we describe an approach for such marginal reconstruction and demonstrate it by reconstructing histories of gene duplications, using common ancestors of ortholog sets as the roots of the integrated-out clades.

We hypothesize that topology features that are robust across many phylogenetic models of divergence are more likely to be accurate and that we can use the variability of topology inference induced by sampling leaves under a set of ancestral nodes of interest to identify the topological features most consistent with robust phylogenetic signal. Testing this hypothesis requires a mechanism for capturing the uncertainty and incorporating it into phylogenetic reconstruction. Inspired by the findings of Salichos and Rokas, who showed that topologies of yeast divergence reconstructed from single genes, while rarely identical, shared much greater similarity than randomly generated topologies (*Salichos and Rokas, 2013*), we propose a new approach based on the possibility that frequently recurring topological features are more likely to represent phylogenetic signal.

In this work, we present a framework for: 1) assessing the amount of uncertainty in the phylogenetic reconstruction of a protein family, 2) identifying topological features and determining the frequency with which they occur across reconstructions, and 3) quantifying the consistency of individual topologies with the identified features. Based on our findings, we propose an observable metric of reconstruction uncertainty, precision, which directly correlates with reconstruction accuracy. We then describe a meta-algorithm that identifies the topologies which are most consistent with topological features extracted from an ensemble of individual reconstructions. Finally, we demonstrate that for some reconstruction tasks, multiple models should be considered as equally likely, given the evidence. Our methodology, ASPEN (Accuracy through Subsampling of Protein EvolutioN), produces significantly more accurate topology models than those produced by a single-alignment reconstruction from all available sequences.

## Results

### Quantifying reconstruction uncertainty with sequence subsampling

Defining protein families for testing

We tested the effectiveness of our analysis and inference framework on simulated protein families and the LacI family of bacterial transcription factors. We turned to simulated sequence data, in addition to a real protein family, for two previously noted reasons (*Wang et al., 2011*). First, simulating evolution over known phylogenies allowed us make a quantitative assessment of reconstruction accuracy compared to the true divergence topology. Second, it allowed us to explore a range of divergence conditions by systematically varying branch lengths of input phylogenies, while controlling for other factors such as overall sequence length and the distribution of secondary structure elements and disordered loops. We simulated 600 protein families, each containing 15 paralogs represented by 66 orthologs – a total of 990 sequences. The scale of this simulation experiment allows for a broad exploration of the effects of topology and divergence time on the accuracy of phylogenetic reconstruction. For simplicity, we used the same tree with 66 species to represent each paralog, resulting in topologies reflecting a series of duplications, producing 15 paralogs in an ancestral genome, followed by speciations leading to 66 divergent genomes, each containing the same 15 paralogs.

Evolution of real protein families is rarely that neat. Speciations can predate duplications. Loss of some paralogs in some lineages can result in different collections of paralogs appearing in different genomes, dramatically complicating orthology assignments. As a result, it can be difficult to determine with a high degree of confidence, whether the leaf sequences in a clade are strictly one-to-one orthologs, or contain some paralogs. In addition, duplication events intermingled with speciation events can produce co-orthologs: many-to-one orthology relationships between genes in post-duplication and pre-duplication species (*Koonin, 2005*). Fortunately, while we demonstrate our methodology using a simplified special case of protein family divergence, the methodology in no way relies on, or benefits from this simplification. As long as high confidence ancestral nodes can be identified, and reconstruction of the clades below those nodes can be sacrificed, the provenance of the integrated-out clades is immaterial. Therefore, the analysis based on this simulation experiment is applicable without loss of generality.

### Framework for generating an ensemble of topologies and quantifying uncertainty

The framework for generating reduced topologies and quantifying reconstruction uncertainty is outlined in *Figure 1*. Full topologies are reconstructed from sequence data by a combination of alignment and phylogenetic inference methods. We then extract ancestor divergence topologies, with the proposal that branch lengths can be optimized later for any combination of selected topology or collection of topologies and alignment of input sequences. Given the framework of our simulated evolution experiment, we focus on the divergence through duplication of common ancestors of orthologs, dispensing with the divergence of the individual orthologs through speciation. In principle, any ancestor of a subset of input sequences can be used, so long as that ancestor is robustly inferred under reconstruction uncertainty, including uncertainty due to input sequence selection and alignment. In our subsampling framework we define robustness as consistency of ancestor reconstruction across subsamples. In other words, most samples from a set of descendants must be monophyletic in their respective reconstructions in order to consider their common ancestor robustly inferred. We found that subsamples from all our simulated orthosets were overwhelmingly monophyletic, so our selection of ortholog common ancestors as leaves for our reconstruction was justified.

We quantified topology differences using the Robinson-Foulds symmetric distance metric (*Robinson and Foulds, 1981*), modified to handle the occasional non-monophyletic reconstruction of an orthoset ancestor ($RF^*$, Materials and methods). When an inferred topology for a synthetic family is compared to the topology over which the family was simulated, that comparison is a measure of the distance between reconstruction and truth. Therefore, we define the accuracy of a reconstruction as:

$$\text{accuracy} = 1 - RF^*(\text{true, reconstructed})$$

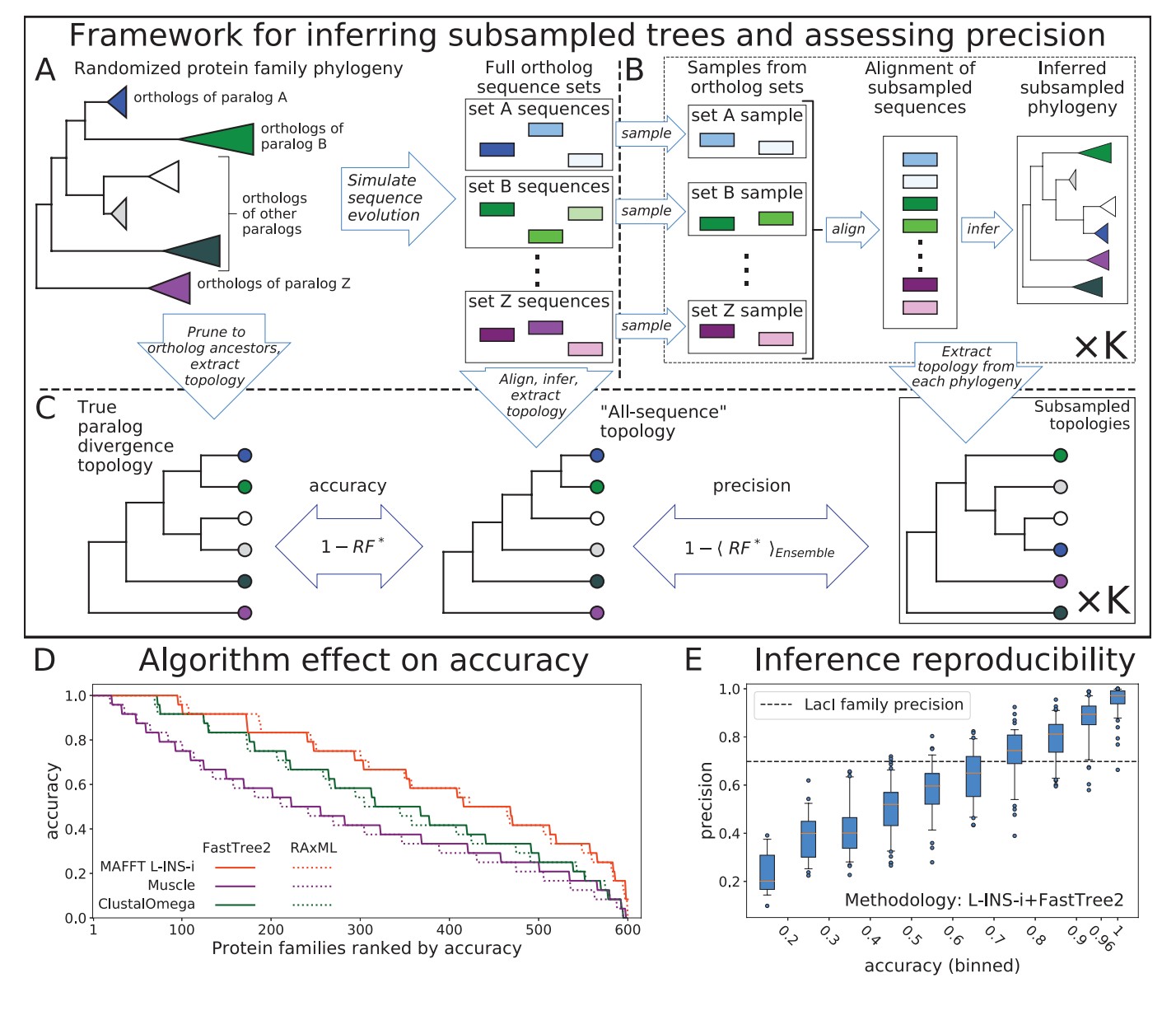

**Figure 1.** Top: Framework for assessing accuracy and precision of reconstruction for simulated protein families using an ensemble of subsampled topologies. (**A**) Sequence evolution was simulated over synthetic phylogenies to produce ortholog sequences (colored boxes), presenting sets of ortholog sequences ("orthosets"). (**B**) Sequences were sampled from orthosets without replacement, aligned, and used to infer "subsampled" phylogenies. (**C**) An all-sequence phylogeny was inferred from an alignment of all sequences in a simulated family. The true, all-sequence, and subsampled phylogenies were pruned to orthoset common ancestors. Their branch lengths were discarded to obtain paralog divergence topologies. Modified Robinson-Foulds ($RF^*$) symmetric distance metric was calculated between the true and all-sequence topologies and between the all-sequence and each subsampled topology. Accuracy and precision of reconstruction for a family are defined in terms of these $RF^*$ distances. Bottom: (**D**) The 600 simulated families are ranked by their all-sequence topology accuracy and plotted according to the alignment and phylogeny inference algorithms used to infer the all-sequence phylogeny. (**E**) Reconstruction precision vs. accuracy of the all sequence topology for simulated families Pearson correlation coefficient between precision and accuracy is 0.91 (p-value < 1e-100). Families are binned by accuracy. Tick marks on x-axis indicate bin boundaries. Dashed line indicates precision of LacI family reconstruction.

DOI: https://doi.org/10.7554/eLife.47676.002

The following source data and figure supplements are available for figure 1:

**Source data 1.** Precision and Accuracy for synthetic protein families.

DOI: https://doi.org/10.7554/eLife.47676.005

*Figure 1 continued*

**Source data 2.** Accuracy of synthetic families for all alignment and inference procedures.
DOI: https://doi.org/10.7554/eLife.47676.006
**Figure supplement 1.** Distributions of pairwise $RF^*$ differences between LacI topologies in the 90% (green) and 50% (blue) subsampling ensembles.
DOI: https://doi.org/10.7554/eLife.47676.004
**Figure supplement 1—source data 1.** Pairwise RF* distributions of LacI topologies.
DOI: https://doi.org/10.7554/eLife.47676.003

We refer to a topology inferred from an alignment of all available sequences as the "all-sequence'' topology. This is the topology most often used when considering a single reconstruction. Using multiple representative sequence sets obtained through subsampling allows us to characterize the uncertainty of ancestor divergence reconstruction arising from selection and, critically, alignment of descendant sequences. In this work, we used two different subsampling sizes to characterize uncertainty. First, we subsampled most, but not all, of the available sequences. The comparisons of these topologies to the all-sequence topology are effectively a measure of the reproducibility of the all-sequence reconstruction given highly similar representations of descendant divergence, but under uncertainty due to input sequence alignment. We refer to this quantity as precision, defined as:

$$\text{precision} = 1 - \left\langle RF^*(\text{all} - \text{sequence}, \text{subsampled}) \right\rangle_{\text{Ensemble}}$$

where angle brackets denote average over the ensemble of subsampled topologies. Second, we subsampled roughly half of the sequences from each orthoset to generate larger ensembles, which we used to identify topological features and quantify their consistency with the phylogenetic signal present in the sequence data. Smaller subsamples allow for faster reconstruction of each topology, in turn allowing us to generate larger ensembles, which provide greater resolution of consistency with phylogenetic signal. The distributions of pairwise $RF^*$ distances between topologies inferred with 90% and 50% subsampling tend to be centered in the same intermediate range as that observed by Salichos and Rokas for species divergence reconstructions from single yeast genes (*Salichos and Rokas, 2013*), with the 50% ensemble allowing for a moderately broader exploration of the topology space (*Figure 1—figure supplement 1*). We settled on this sample size as a compromise between strength of phylogenetic signal, breadth of topology space exploration, and inference speed.

## Selection of alignment and phylogeny inference algorithms
Since the specific alignment and phylogenetic inference algorithms used affect reconstruction (*Morrison and Ellis, 1997*; *Mugridge et al., 2000*; *Wong et al., 2008*; *Wang et al., 2011*; *Liu et al., 2010*; *Liu et al., 2011*; *Blackburne et al., 2013*), we tested all combinations of three alignment algorithms and two phylogenetic inference algorithms on the 600 simulated families, each containing 990 sequences. We used MAFFT's L-INS-i protocol (*Katoh and Standley, 2013*), ClustalOmega (*Sievers et al., 2011*), and Muscle (*Edgar, 2004*) for sequence alignment, and FastTree2 (*Price et al., 2010*) and RAxML (*Stamatakis, 2014*) to infer phylogenies. We compared each all-sequence topology to the true topology and found a wide diversity of accuracies across the 600 synthetic families: accuracy varied between 1, when the all-sequence topology is identical to the true topology, and 0, when $RF^*$ between the two topologies is maximal. Therefore, our test set of simulated families spans a range of difficulty, as observed by a range of inaccuracies across the families, which allows us to observe the performance of phylogenetic reconstruction as a function of that difficulty. The largest effect on accuracy came from the choice of alignment algorithm, consistent with previous studies (*Morrison and Ellis, 1997*; *Mugridge et al., 2000*; *Wong et al., 2008*; *Wang et al., 2011*; *Liu et al., 2010*; *Liu et al., 2011*; *Blackburne et al., 2013*). L-INS-i alignments produced the most accurate reconstructions (*Figure 1D*). FastTree2 and RAxML performed very similarly across all alignment algorithms, also consistent with earlier observations (*Liu et al., 2011*).

Because FastTree2 is significantly faster than RAxML, we selected the fast and accurate combination of L-INS-i and FastTree2 for all subsequent phylogeny inferences.

## Subsampling produces an observable measure of accuracy

We made a striking observation when we compared the reproducibility of an all-sequence tree with its accuracy. Reproducibility, expressed as precision, was measured by comparing 50 trees created from ≈90% subsamples of available sequences (*Figure 1B,C*). As shown in *Figure 1E*, precision directly correlates with accuracy (Pearson correlation coefficient of 0.91 and p-value < 1e-100) – the most accurate all-sequence trees are also the most reproducible. Precision of reconstruction for the LacI family is 0.698, indicated by a dashed line in *Figure 1E*, suggesting that the reconstruction is of moderate difficulty and that the all-sequence LacI topology is not entirely accurate, in accordance with its rather low reproducibility. Due to their strong correlation, it is possible that precision – an observable quantity regardless of whether the true topology is known – can be used as a measure of the difficulty of reconstruction for a protein family and as a proxy for the accuracy of the all-sequence tree.

We wished to understand why subsampling produces a metric that is correlated with accuracy, a hidden value, and to understand the potential generalizability of this observation. To explore this further, we compared the values of the topology search objective function of topologies (log-likelihoods), given an alignment:

$$L(\text{topology}) = -log(\mathscr{L}(\text{phylogeny}|\text{alignment}))$$

by calculating the fractional log-likelihood difference between a reference topology and an alternate topology according to:

$$\text{Difference} = \frac{L(\text{alternative}) - L(\text{reference})}{L(\text{reference})}$$

where the reference is the topology selected by FastTree2 for the alignment in question. We used RAxML to evaluate exact likelihood, rather than the approximate likelihood used by FastTree2, because FastTree2 does not provide the functionality to evaluate its objective function without performing a topology search. We systematically compared topologies on alignments using this fractional difference in log-likelihood to effectively measure the distance between models along the likelihood landscape defined by a specific alignment.

The most immediate question we considered was whether the topology search algorithm simply failed to identify optimal topology models under its own objective function. In particular, we wondered whether this is the reason why the all-sequence alignment failed to recover the true topology for the vast majority of our synthetic families (84.2%). We compared the topologies selected by FastTree2 for all-sequence alignments (the reference topology in our formulation) to true topologies using the log-likelihood difference metric over the all-sequence alignments. A negative value would indicate that the alternate (true) topology obtained a better log-likelihood score than the reference, but the search algorithm failed to identify it. We found no such cases among our synthetic families (*Figure 2A*). While it is reassuring that topology search reliably identified the better scoring topologies, this finding demonstrates that imperfect accuracy results from the fact that, under the substitution model, the observed differences between input sequences are more likely to have arisen by a sequence of substitutions different from the one that actually occurred when evolution was simulated. Even more surprising, the all-sequence topology scored better both under the substitution model used in phylogenetic inference *and* under the different model used for simulating sequence evolution (*Figure 2—figure supplement 1*, and see Materials and methods for choice of substitution models). In fact, for high-precision families, the all-sequence topology scores better than the true topology, given the combination of the substitution model used in simulation and the true alignment. These findings underscore the difficulty of recovering the true topology by phylogenetic inference from short sequences, even using the substitution model under which those sequences were simulated.

Next, we sought to understand how reconstruction of subsampled topologies is related to the accuracy of all-sequence topologies. For a given alignment, the likelihood landscape is a hypothetical multidimensional surface produced by evaluating the likelihood function on topologies, over

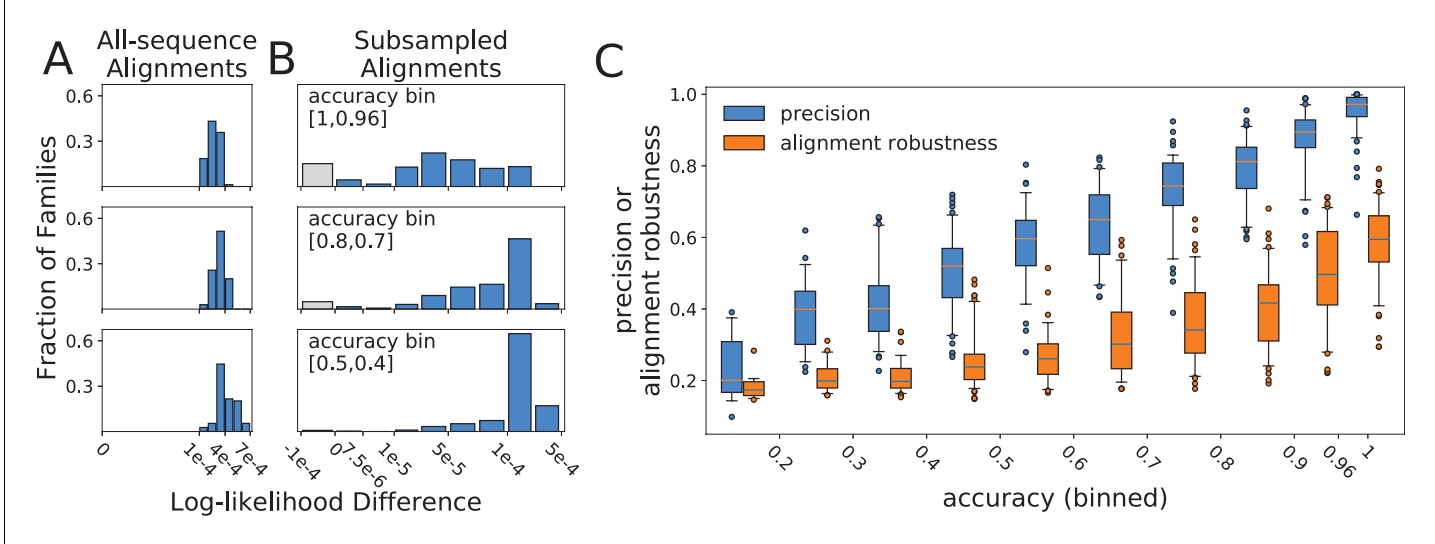

**Figure 2.** Histograms of log-likelihood differences (calculated using RAxML) between true and all-sequence (reference) topologies, calculated over all-sequence alignments (A), and between all-sequence and subsampled (reference) topologies, calculated over subsampled alignments (B). Rows present subsampled and all-sequence results for different accuracy bins. Gray bars represent negative difference values in cases where the alternative topology has a higher RAxML log-likelihood than the topology inferred with FastTree2. These small differences are more likely due to differences between the RAxML and FastTree2 likelihood functions, rather than to topology search failures by FastTree2. (C) Relationship between alignment robustness, quantified by the Guidance2 sum-of-pairs score, and topology accuracy. Alignment accuracy and alignment robustness are 97% correlated, so the relationships of both quantities with topology accuracy are nearly identical. Topology precision is also plotted vs topology accuracy for comparison. Alignment robustness and topology accuracy have a Pearson correlation coefficient of 0.78, while topology precision and topology accuracy have a Pearson correlation coefficient of 0.91.

DOI: https://doi.org/10.7554/eLife.47676.007

The following source data and figure supplements are available for figure 2:

**Source data 1.** Precision and Alignment Robustness data for all protein families.

DOI: https://doi.org/10.7554/eLife.47676.010

**Figure supplement 1.** Histograms of log-likelihood differences between true and all-sequence (reference) topologies, calculated over all-sequence alignments under the JTT (top panel) and WAG (bottom panel) substitution models.

DOI: https://doi.org/10.7554/eLife.47676.009

**Figure supplement 1—source data 1.** Log-likelihood differences between true and all-sequence topologies.

DOI: https://doi.org/10.7554/eLife.47676.008

which the search algorithm seeks the optimal topology for that alignment. We hypothesized that altering the alignment by subsampling sequences and realigning modulates the landscape, relocating the likelihood minima to new topologies, and that the susceptibility of the landscape to such perturbations is related to the accuracy of individual reconstructions. For each subsampled alignment, we compared the subsampled (reference) topology to the all-sequence (alternate) topology using the log-likelihood difference metric. In order to calculate the likelihood of the all-sequence topology given a subsampled alignment, the all-sequence topology was pruned down to the set of leaves contained in the subsampled alignment. Unlike the all-sequence vs. true topology comparisons (*Figure 2A*), the extent of likelihood differences between subsampled and all-sequence topologies differed considerably with accuracy (*Figure 2B*). For families with high accuracy, when precision is also high, the difference varied from 0 to $10^{-4}$, suggesting that the all-sequence topology is within the optimum well. As accuracy and precision decrease, and $RF^*$ distances between subsampled and all-sequence topologies increase, the likelihood differences also increase. The subsampling perturbation produces a greater displacement of the optimum, pushing the all-sequence topology out of the optimum well and onto a plateau, where log-likelihood scores are $10^{-4}$ to $10^{-3}$ worse than the optimum. Greater displacement of the optimum pushes the all-sequence topology out of the optimum well for more synthetic families, but does not increase their log-likelihood differences from the

reference beyond the $10^{-4} - 10^{-3}$ range. We were surprised to discover that true topologies were always located on this plateau, outside of the optimum well on the all-sequence alignment likelihood surface (*Figure 2A*). We suggest that the degree of optimum displacement due to resampling and realignment, captured by our precision metric, reflects the overall dispersion of plausible topology models around the true topology. This dispersion of plausible models is the result of each family's unique divergence history, and both accuracy and precision depend on its extent, explaining why the two are strongly correlated. In other words, less accurate all-sequence topologies are inferred over surfaces more susceptible to the perturbation of input sequence selection and alignment.

Given that subsampling produces new likelihood surfaces determined by the alignments of sub-sampled sequences, we wished to ask whether topology reconstruction accuracy is determined by the accuracy of the input sequences alignment (compared to the true alignment produced through simulated evolution), and whether precision of topology reconstruction is determined by the robustness with which the input sequences can be aligned. If this were the case, then alignment accuracy and robustness would explain the observed precision-accuracy relationship for topology reconstruction. To test this, we measured the similarity between all-sequence and true alignments using the Guidance sum-of-pairs alignment metric (*Penn et al., 2010*), as implemented in Guidance2 software (*Sela et al., 2015*), and compared this to the accuracy of the topology inferred from that all-sequence alignment. Alignment accuracy is correlated with topology accuracy (Pearson correlation coefficient of 0.80), but this correlation is weaker than that between topology precision and accuracy (Pearson correlation coefficient of 0.91).

Next, we asked whether the reproducibility (robustness) of the all-sequence alignment could explain reproducibility of the inferred topology (precision). We used Guidance2 (*Sela et al., 2015*) to determine alignment robustness and found that this is almost perfectly correlated with alignment accuracy (Pearson correlation coefficient of 0.97), but captures slightly less variance in topology accuracy (Pearson correlation coefficient between alignment robustness and topology accuracy of 0.78, *Figure 2C*). This correlation is substantially weaker than the correlation between topology precision and accuracy, driven by the extreme variability in alignment robustness when topology accuracy is high, as well as compression in the scale, compared to precision: no alignment is fully robust, yet some yield completely accurate topologies. Moreover, most of the intermediate accuracy/intermediate precision topologies are inferred from minimally robust alignments (Guidance2 sum-of-pairs score < 0.4). Hence, one could rely on alignment robustness to predict the accuracy of the all-sequence alignment (also a hidden quantity) to a great extent, but that value does not reflect the accuracy of the inferred topology as well as precision of topology inference reflects topology accuracy. We believe this discrepancy is due to the fact that topology selection does not depend equally on all aligned sites (alignment columns), although identifying which sites contribute substantially is far from trivial. On the other hand, the precision of topology inference naturally incorporates alignment robustness to perturbations arising from sequence subsampling, but also incorporates the effect of alignment robustness (or lack thereof) on topology inference, resulting in a metric that is more highly correlated with topology accuracy.

## Exploiting reconstruction variance to identify best trees

Next, we sought to apply our sequence subsampling framework to characterize the region of topology space throughout which subsampled topologies are dispersed, and then to use this information to reconstruct topologies that are most consistent with observations across the subsampled ensemble. We arrived at a reconstruction approach that has three parts: (1) a framework to characterize features observed across an ensemble and quantify their occurrence, (2) a scoring function that reflects the consistency of a given topology with observations across the ensemble, and (3) an algorithm that exhaustively enumerates the highest-scoring topologies according to this metric. We call this approach ASPEN, for Accuracy through Subsampling of Protein EvolutioN.

### Characterizing topological features across an ensemble

First, we required a way of representing topological features observed across multiple topologies and of quantifying the occurrence of those features. Since there is no obvious way to do this using the standard acyclic graph topology representation, we turned to an alternative representation (*Figure 3A*) in terms of the number of internal nodes along the paths between every pair of leaf

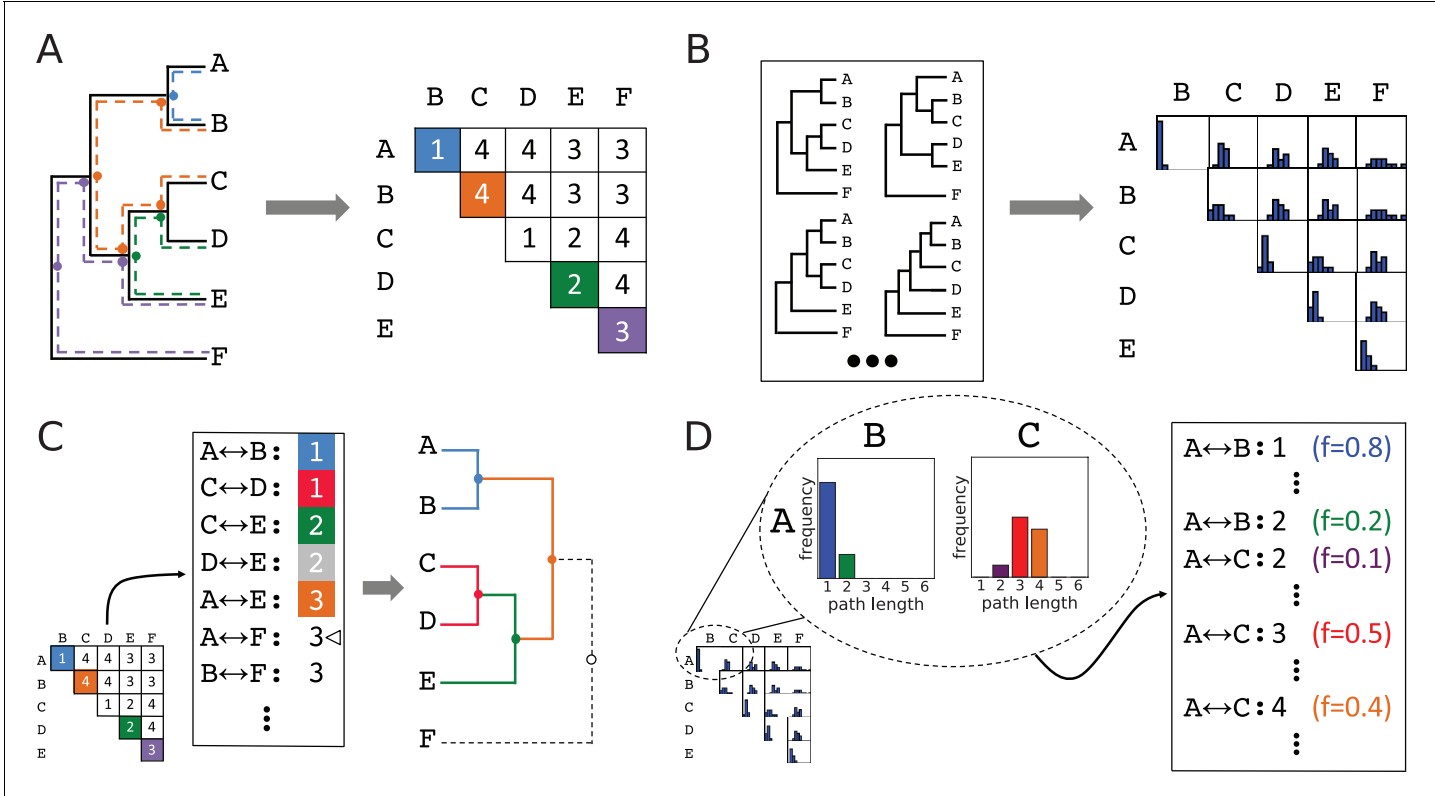

**Figure 3.** Method for accumulating distributions of observations by interconverting between topologies and leaf-leaf distance matrices. (**A**) Conversion of a topology into a matrix of leaf-to-leaf path lengths. Sample paths $(A \leftrightarrow B, 1)$, blue, $(D \leftrightarrow E, 2)$, green, $(E \leftrightarrow F, 3)$, violet, and $(B \leftrightarrow C, 4)$, orange, are highlighted. Dots indicate internal nodes along path. (**B**) Each topology in an ensemble is decomposed into a matrix of leaf-to-leaf path lengths. Observed path lengths for each leaf pair are aggregated into distributions, which are used as weights by ASPEN's log-frequency scoring function. (**C**) Construction of a topology from its matrix of path lengths representation. First, the matrix is transformed into a sorted list of path lengths. Construction of internal nodes is triggered by path lengths encountered traversing the list. Cursor indicates path $(A \leftrightarrow B, 1)$ being recapitulated by construction of node $\{\{\{A, B\}, \{\{C, D\}, E\}\}, F\}$. (**D**) The path lengths distribution for each leaf pair produces as many list entries as there were path lengths observed between those leaves. Each path length for each leaf pair has a corresponding observation frequency. These frequencies are used in the scoring function to rank reconstructed topologies.

DOI: https://doi.org/10.7554/eLife.47676.011

The following figure supplement is available for figure 3:

**Figure supplement 1.** Diagram of branching during ASPEN topology reconstruction.
DOI: https://doi.org/10.7554/eLife.47676.012

nodes. The two representations are equivalent (*Figure 3C*), but the pairwise path lengths representation lends itself to aggregating information across topologies in the form of a path length distribution for each leaf pair (*Figure 3B*), representing empirical probabilities of topological features. Although subsampled reconstructions of simulated orthosets were overwhelmingly monophyletic, occasional reconstructions did contain non-monophyletic ortholog arrangements. This violates an underlying assumption of search space decomposition, as well as the true topology of each simulated family. In such cases, we exclude from the distributions the lengths of any paths from the topology in question that are compromised by passing through spurious nodes resulted from non-monophyletic orthoset reconstruction. Paths from the topology which are not compromised in this way are still included. This highlights a mechanism by which poorly selected ancestors may be identified: when one or more of its putative descendants are consistently placed non-monophyletically with respect to other members of its assigned descendant set, that set's ancestor cannot be considered high-confidence.

## Scoring topologies by consistency with identified features

We can use the frequencies with which specific lengths of leaf-to-leaf paths present in a topology occur in an ensemble to reflect the consistency of that topology with observations across the ensemble, and to make comparisons between proposed topologies. ASPEN formalizes this into a scoring function expressed in terms of log-frequencies of leaf-to-leaf path lengths, $log\left(f_{pair}^{L}\right)$, by summing over all pairs of leaves in the topology, according to:

$$\text{score} = \sum_{\substack{leaf \\ pairs}} log\left(f_{pair}^{L}\right) \tag{1}$$

This scoring function rewards incorporation of frequently observed path lengths and penalizes rarely observed ones.

## Algorithm for constructing N-best trees

Given that topologies inferred from all-sequence alignments tend to be inaccurate, increasingly so for more difficult phylogenetic inference problems (*Figure 1D and E*), ASPEN attempts to identify all likely models of divergence. Specifically, ASPEN's objective is to identify a set of *N*-best topologies that are the most consistent topologies with observations across the ensemble. Given that objective, we created a branch-and-bound procedure to identify the top *N* topologies discussed here. A detailed description of the algorithm is available in Materials and methods. Briefly, branching occurs when a partially constructed topology can be extended by multiple internal nodes. An internal node is permitted as an extension only if every pairwise path completed by the proposed node (*Figure 3C*) appears among the observed path lengths on the list derived from the matrix representation of the subsampled topology ensemble (*Figure 3D*). Every possible extension is realized in a separate extended topology. Construction of internal nodes is triggered to recapitulate path lengths encountered in traversing the list. Since the number of topologies that might be constructed by this branching can be very large, even given the constraints of ensemble observations, we use bounding to limit construction to the *N* best-scoring topologies. Bounding occurs by checking whether a partially constructed topology might be completed with a better score than the current *N*th-best completed topology. If this is not possible, the partially constructed topology is discarded, bounding all branched construction paths by which it could have been extended. Upon completion of the branch-and-bound procedure, ASPEN will have identified and ranked the *N*-best topologies, according to their consistency with observations from the ensemble of topologies created by subsampling available sequences.

## Evaluation of ASPEN reconstructions

To test the accuracy of ASPEN reconstructions, we used the outlined framework (*Figure 1*) to generate an ensemble of 1000 subsampled topologies. For this ensemble, we subsampled 30 of 66 orthologs of each paralog in the synthetic families ($\approx 45\%$) and reconstructed the best 10,000 topologies for 400 families. Here, we consider the top 50 topologies constructed by ASPEN for each family, ranked according to their log-likelihood score, reflecting their consistency with the ensemble.

## ASPEN's top topology is more accurate

First, we explored how the top-ranked ASPEN topology compares to the the all-sequence topology inferred using the most accurate combination of algorithms (MAFFT L-INS-i alignment and FastTree2 phylogeny inference). *Figure 4A* demonstrates that in the vast majority of cases (90.7% of all families) the top-ranked ASPEN topology is equal to or more accurate than the all-sequence topology. The ASPEN topology is more accurate for 53.8% of families. At the highest precision values, where all-sequence topologies are the most accurate, the top-ranked ASPEN topology is the true topology. ASPEN provides greatest improvement over single-alignment inference when precision is below 0.9, where it is very likely that ASPEN's top topology will be more accurate than the all-sequence topology. In general, top-ranked ASPEN topologies for low precision families are also not completely accurate. However, there are some cases where medium/low accuracy of the all-sequence topology corresponds to a completely accurate top ASPEN topology. Therefore, ASPEN's

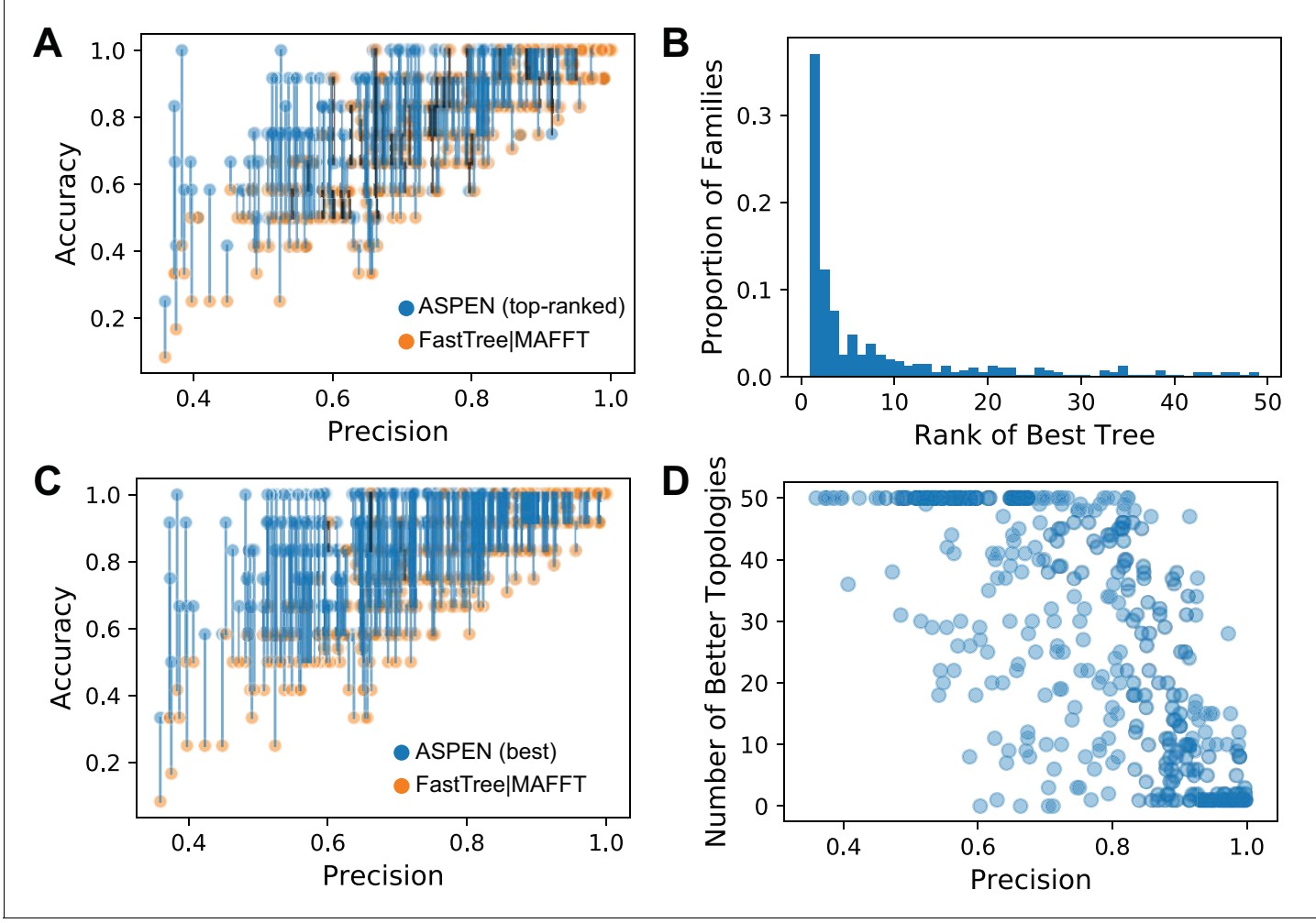

**Figure 4.** The accuracy of ASPEN models. (**A**) Comparison of the accuracies of the all-sequence and top-ranked ASPEN topologies, arranged according to families' precision values. The two accuracy values for each family are connected by a blue line if the ASPEN topology is more accurate than the all-sequence topology, by a black line if the all-sequence topology is more accurate, and no line if their accuracies are equal. (**B**) A histogram reflecting where the most accurate ASPEN topology falls among the top 50 ASPEN topologies for that family. (**C**) Comparison of the accuracies of the all-sequence and the *most accurate* topology among the top 50 of ASPEN topologies. Connections between the points have the same meaning as panel A. (**D**) The number of top-50 ASPEN topologies that are more accurate than the all-sequence topology for that family. Each circle represents a synthetic family.

DOI: https://doi.org/10.7554/eLife.47676.013

The following source data is available for figure 4:

**Source data 1.** Accuracy, Precision, and Rankings of ASPEN topologies.
DOI: https://doi.org/10.7554/eLife.47676.014

---

top topology provides the greatest accuracy improvement over single-topology reconstructions when reconstruction is most complex.

## Log-frequency score is related to accuracy

Next, we wished to explore whether ASPEN's ranking system, based on likelihood according to the ensemble, identifies more accurate trees in general. *Figure 4B* captures where the most accurate ASPEN topology among the top 50 ASPEN topologies is ranked according to the ASPEN likelihood scoring function. The fast drop-off in rank demonstrates that the most accurate topologies are ranked highest by ASPEN. The top-ranked topology is also the most accurate for 36.9% of families,

while for the remaining 63.1% of families the most accurate tree is not the top-ranked tree. However, the most-accurate tree recovered is still related to the topologies likelihood score and 75% of the time the most accurate tree is in the top ten trees returned by ASPEN. Together, these results demonstrate that ASPEN's approach and likelihood scoring function, which reflect consistency with ensemble-observed features (*Equation 1*) are related to topology accuracy.

## ASPEN produces many topologies with better accuracy

*Figure 4C* demonstrates how much better the most accurate topology found among ASPEN's top 50 is than the all-sequence topology. For 99% of families, the most accurate among the top 50 topologies returned by ASPEN is better or equal in accuracy to the all-sequence topology (76% of the time it is better). Hence, the top-ranked topology is generally better than the all-sequence topology (*Figure 4A*), but there are bigger gains for topology improvement when considering other topologies within the ASPEN top-50. In order to understand how many topologies are better, we calculated the fraction of the 50-best ASPEN trees that are better than the all-sequence topology (*Figure 4D*). Not surprisingly, for families with highly accurate all-sequence topologies, which also have high precision, one or few topologies are equally or more accurate, which corresponds with the top-ranked ASPEN tree also recovering the true topology. Strikingly, as precision drops, even a small amount, the number of topologies that are more accurate than all-sequence increases. At the extreme-end, where precision is at its lowest observed, all 50 ASPEN topologies are more accurate than the all-sequence topology.

## How ASPEN produces more accurate topologies

We wanted to understand how log-frequency scoring facilitates identification of more accurate topologies than phylogenetic reconstructions from single alignments. Using the length of path between two leaves, we explored the connection between path length frequencies observed across an ensemble and differences between true, all-sequence, and ASPEN topologies.

First, we compared the observation frequencies of path lengths between true and all-sequence topologies. Among paths on which the two topologies disagree, the length consistent with the true topology was observed more frequently than the length consistent with the all-sequence topology for half or more paths across all but the highest-precision bin (*Figure 5A*). In the highest-precision bin, reconstruction is extremely accurate and the true and all-sequence topologies disagree on a very small fraction of path lengths (4.8%). Although the fraction of *all* paths with incorrect length in the all-sequence topology (overall bar height in *Figure 5A*) increases dramatically as precision falls, the fraction of *disagreeing* paths for which the ensemble correctly identifies the true length (fraction of bar filled with blue) remains surprisingly constant. The larger fraction of all paths for which the ensemble supports the true path length (height of blue bar segment) accounts for the much slower drop off in accuracy of all ASPEN topologies at lower precision, compared to the precipitous fall in the accuracy of all-sequence topologies. Unfortunately, the fraction of all paths for which the ensemble supports the incorrect all-sequence path length (height of white bar segment) also increases at lower precision, which explains why any drop off in ASPEN topology accuracy occurs at all. Nevertheless, the frequency of the most frequent path length and the breadth of the path length distribution provides a measure of confidence in the ensemble's support of a particular path length. The aggregate of this confidence across all pairwise paths is reflected in the log-frequency score differences between ASPEN topologies.

Next we examined the agreement, in terms of path lengths, between the top ASPEN topology, the true topology, and the all-sequence topology (*Figure 5B*). As expected, the fraction of all paths with an incorrect length in one or both of the all-sequence and top ASPEN topologies (overall bar height in *Figure 5B*) is larger at lower precision. Surprisingly, the fractions of all paths with the correct path length in the top ASPEN topology (height of blue bar segment) and the all-sequence path length in the top ASPEN topology (height of empty bar segment) change little as precision falls, while the fraction of all paths with lengths in the top ASPEN topology matching neither the true length nor the all-sequence length (unique paths, height of gray bar segment) increases. This discrepancy may explain why the accuracy difference between the top ASPEN topology and the other ASPEN topologies decreases at lower precision. Although the fraction of paths with an incorrect length in the all sequence topology, but with the correct length identified through subsampling

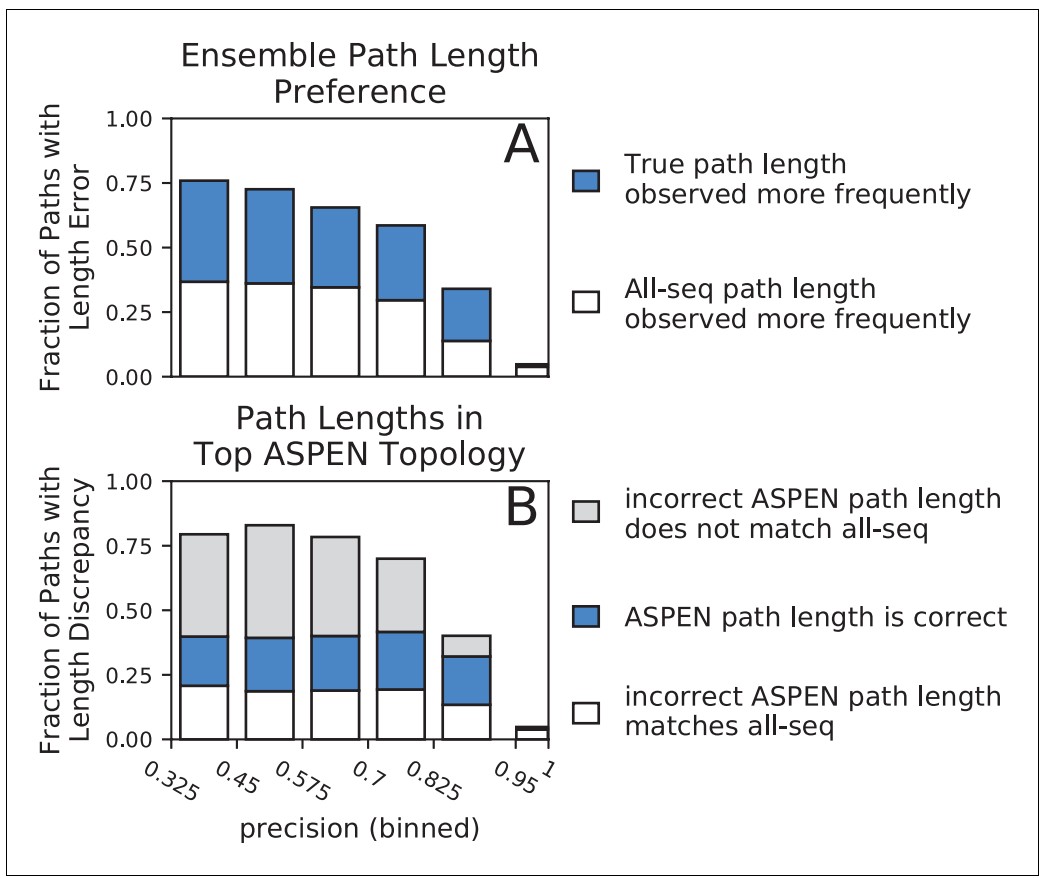

**Figure 5.** Accuracy of path lengths in the subsampled ensemble and the top ASPEN topology. Synthetic families were binned by precision and path length data was aggregated from all families in a precision bin. (**A**) Total height of bar represents fraction of all paths across all families in precision bin with a length error (path length is incorrect in all-sequence topology). Blue fraction of bar represents paths for which the true length was observed more frequently across the ensemble. White fraction of bar represents paths for which the incorrect all-sequence length was observed more frequently across the ensemble. (**B**) Total height of bar represents fraction of all paths in bin with a length discrepancy (on which the true, all-sequence, and top ASPEN topology fail to agree). Blue fraction of bar represents paths with the correct length in the top ASPEN topology. White fraction of bar represents paths on the incorrect length of which the top ASPEN topology agrees with the all-sequence topology. Fraction of bar shaded gray represents paths of unique length in the top ASPEN topologies: incorrect length different from the length in the all-sequence topology.

DOI: https://doi.org/10.7554/eLife.47676.015

The following source data is available for figure 5:

**Source data 1.** Path length differences.
DOI: https://doi.org/10.7554/eLife.47676.016

(height of blue bar segment in *Figure 5A*) increases, not all such path lengths are incorporated into the top ASPEN topology – likely due to the constraints imposed by other path lengths on the reconstruction of internal nodes. Instead the correct path lengths are incorporated into other topologies proposed by ASPEN. Accordingly, log-frequency score differences between ASPEN topologies also decrease at lower precision (*Figure 6*, *Figure 6—figure supplement 1*), reflecting more uniform confidence in any individual topology.

## ASPEN reconstruction of LacI paralog divergence

As mentioned previously, LacI falls into an intermediate range of reconstruction difficulty. In this range, the 10 to 30 highest ranked ASPEN topologies are likely to be more accurate than any all-

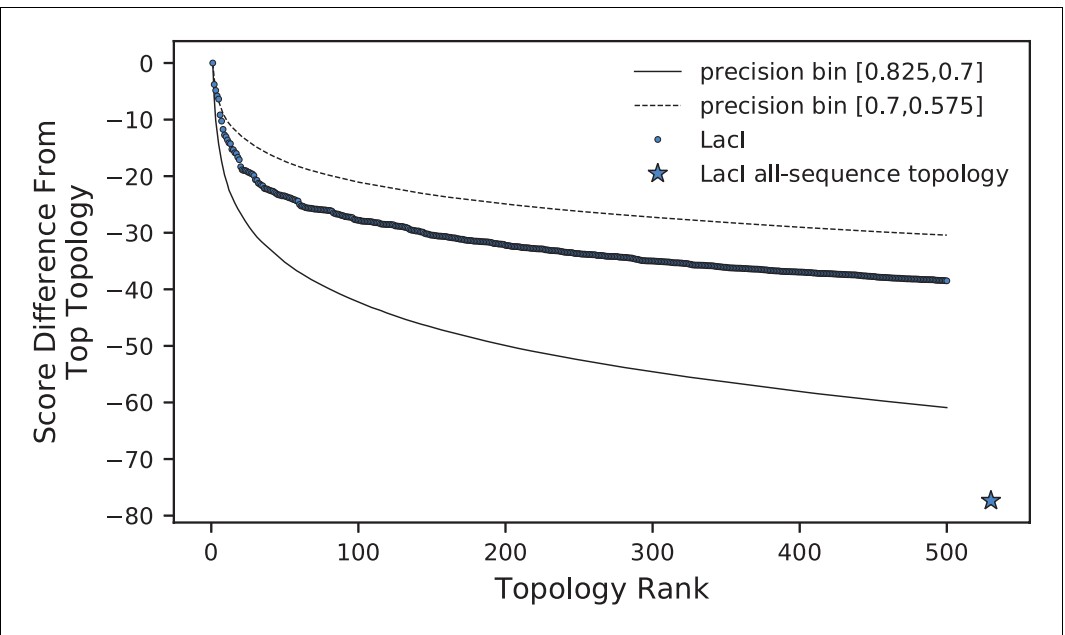

**Figure 6.** Comparison of log-frequency scores of the top 500 ASPEN topologies for LacI. The difference between a topology's score and the score of the best ASPEN topology is plotted as a function of topology rank. Bin-average score differences for simulated families from the two precision bins between which LacI falls are plotted for reference. Also plotted is the difference in score between the all-sequence topology of LacI and the top ASPEN topology.

DOI: https://doi.org/10.7554/eLife.47676.017

The following source data and figure supplement are available for figure 6:

**Source data 1.** Differences in log-frequency score by rank.
DOI: https://doi.org/10.7554/eLife.47676.019

**Figure supplement 1.** Dropoff in ASPEN log-frequency scores across precision bins.
DOI: https://doi.org/10.7554/eLife.47676.018

sequence reconstruction, based on our observations from synthetic protein families (*Figure 4D,E*). Given this, we reconstructed LacI paralog divergence using ASPEN. We derived path length frequencies from an ensemble of 1000 subsampled topologies (*Figure 1B*), using ~50% of the available ortholog sequences (the same procedure that was used for synthetic families). We then used ASPEN to construct the best 500 topologies (Supplementary Material). *Figure 6* plots the drop-off in the log-frequency score of each ASPEN topology, compared to the top-ranked topology, for LacI and for the two precision bins at the boundary of which LacI falls (*Figure 4D,E*). Log-frequency scores decay faster at higher precision (*Figure 6*), reflecting a greater difference in confidence for each lower ranked topology, as previously discussed. The all-sequence LacI topology does not appear among the top 500 ASPEN topologies, having scored significantly worse then the ASPEN trees according to the log-frequency scoring function. This indicates that all 500 ASPEN topologies are more consistent with observations across the ensemble of subsampled LacI topologies than the all-sequence reconstruction.

A comparison of all-sequence and top ASPEN topologies (*Figure 7*) illustrates why the all-sequence topology scores so poorly against the 50% subsampling ensemble. Since the log-frequency scoring function penalizes topologies for incorporating rarely observed leaf-to-leaf path lengths, infrequent incorporation of an ancestral node into ASPEN topologies indicates that most clade arrangements below the node produce unfavorable path lengths. While the top ASPEN topology incorporates the [Mal-B, AscG, GalRS] common ancestor, which appears in and additional 46% of ASPEN topologies, alternative placement of the Mal-B terminal branch in the all-sequence topology produces a different ancestral node, which appears in only 24% of ASPEN topologies. Worse,

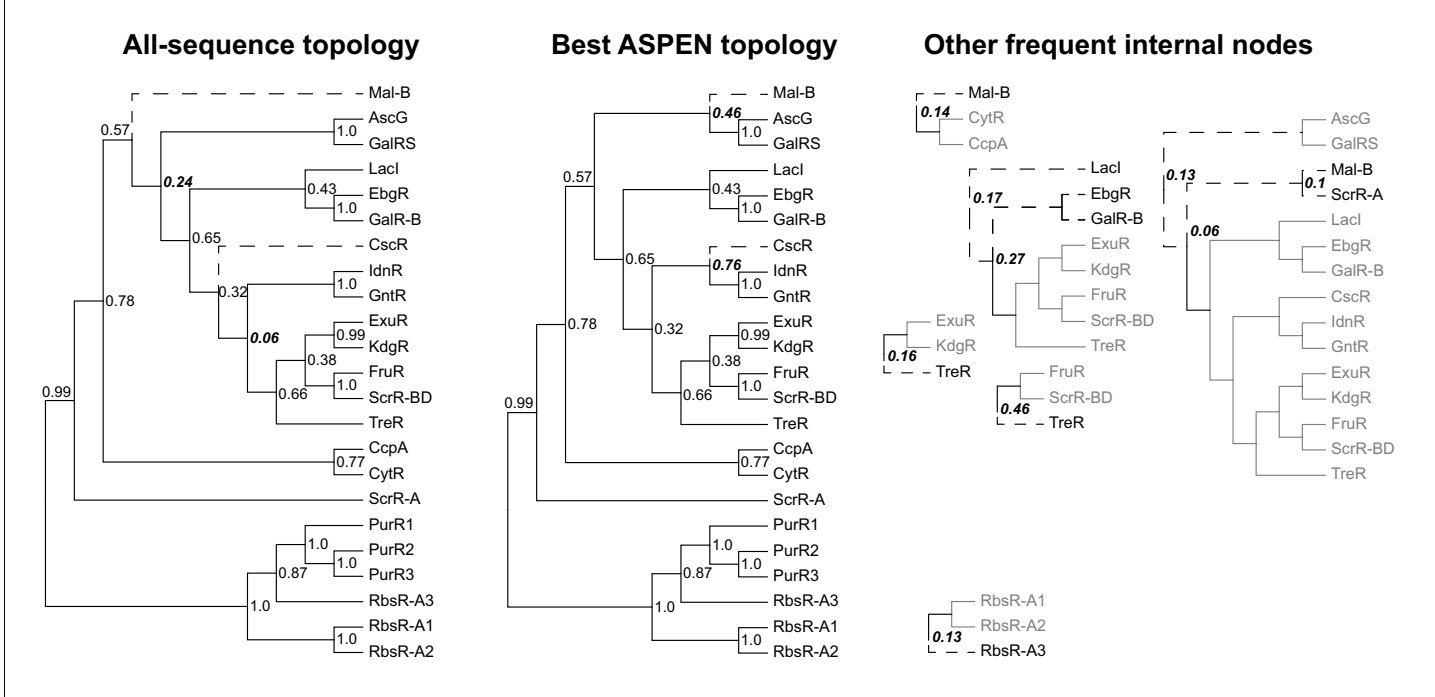

**Figure 7.** Reconstructed topologies for the LacI family. Reconstructed nodes are annotated with the frequencies at which they were recapitulated among the 500 top-scoring topologies reconstructed by ASPEN as a way of summarizing ancestral nodes observed across the most likely trees. Subtrees on right represent reconstructed nodes observed with frequency ≥0.1 among the 500 ASPEN topologies, but not appearing in either the all-sequence or the best ASPEN topology. Branches placed differently in the all-sequence and best ASPEN topologies are shown as dashed lines, as are branches placed differently from either topology in the subtrees on the right. Observation frequencies for disagreeing splits are bolded and italicized.
DOI: https://doi.org/10.7554/eLife.47676.020

the all-sequence topology is missing the [CscR, IdnR, GntR] common ancestor, which appears in 76% of ASPEN topologies, incorporating instead the [IdnR, GntR, ExuR, KdgR, FruR, ScrR-BD, TreR] common ancestor, which appears in only 6% of ASPEN topologies. Taken together with our findings for synthetic families, these results suggest that the best ASPEN topology is more accurate than the all-sequence topology, but that none of the reconstructed topologies are likely to match exactly the true divergence of LacI paralogs. In lieu of using a single topology, downstream analyses would do well to reflect this uncertainty by considering multiple likely topologies produced by ASPEN.

## Discussion

We described a novel approach to analyzing and reconstructing divergence histories of protein families. Our approach is conceptually rooted in the decomposition of the topology search space at high confidence ancestral nodes, which are extremely likely to exist in the true topology, and takes advantage of the fact that complete divergence histories include ''nuisance'' segments, which provide little biological insight. Instead of reconstructing such segments, we propose integrating over the uncertainty of their reconstruction to produce more accurate "marginal'' reconstructions of the most interesting segments. Critically, our approach considers the uncertainty arising from input sequence selection and alignment, a historically thorny issue in phylogenetic analysis (*Wong et al., 2008*). The traditional method of assessing the reliability of phylogenetic reconstruction, the phylogenetic bootstrap (*Felsenstein, 1985*), cannot address the reliability of the individual sites (alignment positions) it resamples. On the other hand, the sequence resampling approach we presented occurs farther upstream in the inference process, treating sequence alignment and phylogeny reconstruction as a single inference procedure subject to multiple sources of uncertainty. We can use the resulting ensemble of subsampled topologies: 1) to compute an observable metric, precision, which is directly

proportional to the accuracy of any individual reconstruction – a hidden quantity for reconstructions from real sequences – and 2) to assemble leaf-to-leaf path length frequency distributions, which we use to define the log-frequency scoring function that is also directly related to reconstruction accuracy. Our topology reconstruction algorithm then uses the scoring function to identify and rank topologies according to their consistency with the phylogenetic signal characterized by these empirical distributions. The highest scoring topologies are more accurate than topologies reconstructed from alignments of all available sequences, confirming that the topological features more frequently represented across subsampled topologies are also more consistent with the true phylogenetic signal. Crucially, ASPEN identifies these topologies in the face of misleading likelihood landscapes resulting from each individual input alignment, on which the true topology is not the maximum likelihood topology, or even located within the optimum well. Finally, we showed that, although ASPEN tree accuracy declines as the reconstruction task gets harder (as evidenced by decreased precision and accuracy of the all-sequence tree), its decline is significantly slower than that of single all-sequence reconstructions. Importantly, ASPEN is able to identify when an all-sequence tree is likely to be inaccurate (via precision) and then construct and rank a set of trees which, while unlikely to be exactly correct, are all likely to be more accurate than any all-sequence tree. We propose that downstream analysis relying on a divergence topology should aim to integrate over this topological uncertainty.

## Generality of the accuracy-precision relationship

Given the potential usefulness of the correlation between accuracy and precision, it remains vitally important to ascertain the robustness of this relationship to varying degrees of mismatch between actual evolutionary processes which gave rise to extant sequences and the evolutionary model used in phylogenetic inference. In our sequence evolution simulations, we rescaled the grafted species tree, both as a whole and its individual segments, to model evolutionary rate differences between paralogs (heterotachy) and used the JTT amino acid substitution model with variable branch length scaling spanning a five-fold range (*Supplementary Material*) to model among-site differences in substitution rate across the protein fold. To incorporate model violation into our phylogenetic reconstructions, we used the WAG substitution model with substitution rate heterogeneity modeled by a discrete gamma distribution and no mechanism to model heterotachy. However, the evolutionary histories of real protein families may be subject to a number of additional forms of model violation, such as among-site variation in amino acid frequencies, varying selection pressures leading to convergent evolution, a known cause of the long branch attraction artifact in phylogenetic reconstruction, and, crucially, non-independent evolution of sites.

Although, in general, positions in protein sequences clearly do not evolve independently of one another, characterizing the sources and effects of positional interdependence in protein evolution has proven difficult (*Arenas et al., 2013*; *Grahnen et al., 2011*; *Choi et al., 2007*; *Rodrigue et al., 2005*; *Robinson et al., 2003*; *Parisi and Echave, 2001*). Unlike structural RNA molecules, where base pairing leads to strong pairwise dependence between sites, protein sites tend to be more weakly dependent on larger numbers of other sites (*Nasrallah et al., 2011*). Accuracy of phylogenetic inference in general appears to be affected in a sequence length-dependent manner, with site non-independence resulting in less phylogenetic signal than site-independent sequences of the same length (*Nasrallah et al., 2011*; *Huelsenbeck and Nielsen, 1999*). The concept of effective sequence length was introduced by *Nasrallah et al. (2011)*, where effective sequence length equals actual sequence length for site-independent sequences, but is shorter than actual length for non-independent sequences, reflecting reduced phylogenetic signal. If reduction in phylogenetic signal is indeed the principal effect of non-independent evolution of sites, then precision should diminish accordingly, and the strength of the accuracy-precision relationship should not be affected.

Given a particular set of evolutionary circumstances, the correlation between accuracy and precision may be diminished under those circumstances in one of two ways: 1) systematically higher accuracy than would typically be indicated by the corresponding degree of precision, or 2) systematically higher precision than the degree of accuracy. The former – *systematically* higher accuracy than that indicated by precision for all protein families evolved under the circumstances in question – implies a critical threshold in the number of input sequences between all-sequence and high-density subsampled reconstructions. Given thorough ortholog representation of each paralog, this seems exceedingly unlikely. The latter – misleadingly high precision – would reflect a systematic bias toward

specific incorrect topologies under those particular evolutionary circumstances. This is exactly the case with long branch attraction, and can plausibly occur in other ways as well. Identifying evolutionary circumstances which result in such systematic breakdown of the accuracy-precision relationship is vitally important for future work.

## Selection of ancestral nodes for topology search space decomposition

In this study, we limited evolutionary simulations to a simplified scenario: divergence of 15 paralogs through duplication events, followed by divergence of orthologs through speciation without any further duplications or gene loss. This produced topologies with the divergence history of 66 species fully recapitulated under the common ancestor of each ortholog set. With only duplication nodes above ortholog common ancestors, and only speciation nodes below them, these 15 ancestors were the natural choice for roots of integrated-out clades in our analysis. Inference of ortholog common ancestors was nearly universally recapitulated across all subsamples for all 600 simulated families, so these nodes satisfied our high confidence criterion for decomposition of the topology search space.

In real protein families, including the LacI family, duplications are typically intermingled with speciations. For distantly related species, where multiple duplications have occurred along both lineages since the species diverged, this can produce collections of paralogs without orthology relationships to each other. For more closely related species, where one or more duplications occurred along one of the two lineages, it can result in co-orthology, mapping multiple genes in a post-duplication genome to a single gene in a genome from the other lineage. In addition, loss of paralogs along some lineages can result in different numbers of paralogs in different species, with unclear orthology vs paralogy relationships. ASPEN analysis of any real protein family starts with selecting the ancestral nodes for search space decomposition. Any collection of internal nodes, regardless of the relationships among leaves within each clade or their relationships to leaves outside the clade, can be selected for this purpose. In our analysis of the LacI family, the leaf sets in integrated-out clades did not come from a uniform collection of species, suggesting that post-speciation duplications and/or gene loss occurred in the evolution of LacI paralogs. The only requirement is sufficient confidence in each ancestor's existence, as evidenced by robust inference over any set of reasonable input alignments or alignment perturbations. In our opinion, ancestors with predominantly speciation nodes below them and predominantly duplication nodes above, as evidenced by leaf sequence species of origin, result in analysis with the most biological insight, but this is not a requirement. Although this step must be carried out individually for any protein family, based on the information researchers wish to obtain from the analysis, we provide some suggestions for how to apply our methodology to one's protein family of interest.

We recommend relying on existing genomic and/or functional annotations and average within-group vs. average out-of-group sequence similarity (separability) to identify candidate leaf subsets to integrate out. When dealing with large families, such as some protein domains, which can number in the dozens or hundreds of paralogs in vertebrate genomes, common ancestors of multiple closely related paralogs may be logical choices. ASPEN methodology can be used recursively to separately reconstruct the divergence of these paralogs – another advantage of decomposing the search space at high confidence nodes. Next, we recommend reconstructing an all-sequence topology and a small ensemble of densely subsampled topologies to assess confidence in the existence of selected ancestors, as well as to determine reconstruction precision and form an expectation of the ultimate accuracy of reconstruction. If selected leaf subsets are not reliably monophyletic across the ensemble, the selections need to be revisited until the common ancestor of each integrated out clade can be inferred with high confidence.

Clearly, applicability of ASPEN analysis to any particular family hinges on the existence of such high confidence ancestors and the willingness to sacrifice reconstruction of clades below those ancestors. In part, this requirement can be satisfied by careful assembly of the input sequence collection: including as many ortholog or co-ortholog sequences as possible will maximize the chances of high confidence reconstruction of the common ancestors of those orthologs, provide large leaf subsets for subsampling, and mitigate the conditions under which the long branch attraction artifact occurs. Nevertheless, it is possible that individual leaf sequences or common ancestors of a few sequences (e.g. five or fewer) will not be robustly monophyletic with any other leaves. We recommend including these leaf sequences in every subsample and treating them (or their common ancestors, for small monophyletic clades) as leaves in ASPEN topology reconstruction. ASPEN analysis

remains applicable given at least one integrated out clade, although we expect the usefulness of that analysis to diminish as more leaf sequences fall outside of such clades.

## Practical considerations for ASPEN topology reconstruction

ASPEN's branch-and-bound algorithm provides a powerful guarantee of completeness – that the *N*-best trees were produced – at the end of its run, but execution times and resource requirements can be substantial for large families. ASPEN's reconstruction is parallelizable across multiple CPUs and we generally observed that the least difficult (most accurate) synthetic families were complete within an hour of total CPU time, whereas the most difficult problems could take up to 400 CPU hr. However, the vast majority of *N*-best trees were always identified early in the the run (within a few hours), with the remainder of the run spent almost exclusively rejecting worse topologies. Dispensing with the branch-and-bound guarantee, topology assembly can be truncated after a fraction, sometimes a very small fraction, of the full run time, retaining a nearly-complete collection of *N*-best trees. Additionally, there are possibilities for other heuristic approaches that could scale better with the inference complexity and the number of nodes than the current implementation of ASPEN that uses the log-likelihood with the ensemble as an objective function, now that it has been shown to be capable of identifying more accurate topologies.

Once ASPEN has produced a collection of likely topologies of ancestor divergence, researchers may want to obtain complete phylogenies for their input sequences by grafting topologies of integrated out clades at the corresponding leaves of ASPEN-reconstructed topologies and optimizing branch lengths for the resulting full topologies. For common ancestors of orthologs, we strongly recommend using the species divergence topology for the divergence of orthologs, unless truly compelling evidence to the contrary exists. In our analysis of simulated families, where ortholog evolution was simulated over the same species tree in each case, the correct speciation topology was never recovered for all 15 paralogs, underscoring the futility of reconstructing species divergence from single protein families. For common ancestors of multiple paralogs, ASPEN can be applied recursively to infer the divergence of those paralogs. After complete topologies have been assembled, branch lengths and other parameters can be optimized for any given sequence alignment, for example using RAxML (*Stamatakis, 2014*).

## Scalability of ASPEN and extensions to the methodology

Traditionally, the effect of alignment uncertainty on phylogenetic reconstruction has been addressed through simultaneous inference of the alignment and the phylogeny, either through extensions of the approach used by progressive multiple sequence alignment algorithms: alternating iterative refinement of alignment and phylogeny (*Mirarab et al., 2015*; *Liu et al., 2012*; *Liu et al., 2009*; *Parmentier et al., 2006*; *Edgar and Sjölander, 2003*), or through full joint co-estimation of alignment and phylogeny by Bayesian Markov Chain Monte Carlo (*Suchard and Redelings, 2006*; *Redelings and Suchard, 2005*). While iterative refinement can handle much larger reconstructions than Bayesian co-estimation (*Mirarab et al., 2015*; *Liu et al., 2012*), both approaches incur the full cost of repeated inference of every ancestral node, including all speciation nodes. Algorithmic advances and improved parallelization (*Mirarab et al., 2015*; *Stamatakis, 2014*; *Katoh and Standley, 2014*) permit larger reconstructions through heuristic approximations, but the computational cost of repeated large phylogenetic reconstructions constitutes a fundamental challenge. Abandoning reconstruction of every ancestral node allows ASPEN topology reconstruction to scale much better. Unlike in simultaneous alignment and phylogeny inference, all of the large phylogenetic reconstructions required to build a topology ensemble can be carried out concurrently, allowing ASPEN to take maximal advantage of high-performance computing resources. ASPEN's topology reconstruction algorithm itself is not affected by the number of leaf sequences within integrated-out clades, only by the number of such clades, making it scale much better with the total number of input sequences than simultaneous co-estimation approaches.

We anticipate that, as a meta analysis approach to tree evaluation and reconstruction, ASPEN is likely to continue to boost the accuracy of individual alignment and tree reconstruction approaches, regardless of the specific underlying alignment and reconstruction algorithms or the application (e.g. whether it is subsampling from orthologs to infer paralog relationships or subsampling from genomes to reconstruct species relationships). Alternate statistical approaches are increasingly

important with the advent of affordable genome sequencing and the resulting explosion in the number of sequenced and annotated species' genomes (*Aken et al., 2017*; *Benson et al., 2017*). The current implementation of ASPEN as a subsampling and scoring approach is immediately tractable for nearly all protein families. ASPEN's branch-and-bound reconstruction algorithm is also immediately tractable for reasonably sized families, such as the LacI family. However, the sequence subsampling approach, the path-length frequency distributions it provides, and the log-frequency scoring function are powerful tools in their own right, which scale with the number of integrated out clades much better than the branch-and-bound algorithm. A more efficient search of topology space under this objective function, with a guarantee and/or estimation of completeness, is possible. It may be possible to structure subsampling so that the resulting frequency distributions are even more reflective of the true topology. Finally, a mechanism for robust inference of branch lengths for ASPEN-constructed topologies that, similar to ASPEN topology reconstruction, integrates over the uncertainty of alignment and reconstruction below selected ancestors, is clearly desirable.

## Conclusions

Phylogenetic inference is an extremely difficult problem, particularly for individual protein families, where short amino acid sequences contain limited phylogenetic signal. Here, we demonstrated that the sensitivity of inference and alignment algorithms to their inputs is a result of the failure of objective functions to accurately model the mutational events that underlie the divergence nodes in an evolutionary tree, which manifest in both alignment and inference procedures. At their core, phylogenetic trees are hypothetical models of a currently unknowable evolutionary history. George E.P. Box's much quoted philosophy about models is relevant here: "Essentially, all models are wrong, but some are useful''. Extending this, we propose that integrating across a lot of models that are wrong in different ways, but right in the same way, yields more useful models of evolutionary divergence. This is based on the observation that topologies capturing more of the shared information across many incorrect models are significantly more accurate. However, we also find that, for especially difficult inference problems (where all-sequence topologies are imprecise, that is have low reproducibility under subsampling) it is important to consider multiple models as equally viable explanations of evolutionary history. In other words, for most real evolutionary inference problems, scientists should integrate their analyses over multiple plausible phylogenetic models. The methodology we presented facilitates both an overall assessment of accuracy of topology reconstruction for any particular problem, and, in light of that assessment, reconstruction of an appropriate number of maximally plausible topological models. Crucially, ASPEN is geared toward analysis of precisely the most difficult class ancestors: ancient ancestors of multiple paralogs.

It should be noted that individual ancestral nodes can vary dramatically in their difficulty of inference, and many ancestors of great biological interest can be inferred with very high accuracy, despite low overall accuracy of topology inference for that family. Moreover, given high confidence in the existence of such an ancestor, reconstruction of that ancestor's sequence is robust with respect to topological uncertainty for the clade below it (*Eick et al., 2017*). A critical step in evolutionary analysis, then, is to determine *which* inferred ancestors are reliable – both with respect to systematic biases and with respect to uncertainty due to various sources of random noise – and which are not. ASPEN topologies can be used for this purpose in a fashion similar to the traditional phylogenetic bootstrap, with the additional benefit of evaluating consistency of an ancestor's inference across the most likely topologies.

Divergence reconstructions for well-studied protein domain families are relied upon extensively by the scientific community. For example, evolutionary trees of catalytic and recognition protein domains involved in signaling, including protein kinases (*Manning et al., 2002*) and phosphatases (*Chen et al., 2017*), SH2 domains (*Liu et al., 2006*), de-ubiquitinating enzymes (deubiquitinases or DUBs) (*Nijman et al., 2005*), histone deacetylases (HDACs) (*Gregoretti et al., 2004*), and Ras GTPases (*Rojas et al., 2012*) are ubiquitously used. Because such reconstructions are created from single sequence alignments, they ignore the great deal of uncertainty in topology reconstruction under equally valid representations of available sequence data. Furthermore, such reconstructions are often built from limited data. For example the human kinome, which has been cited over 6000 times to date, was constructed just from human sequences – an example of extreme subsampling, with each ancestor represented by a single sequence. Topology reconstructions from single alignments with sparse subsampling are likely to be extremely inaccurate. We found individual

reconstructions are extremely unreliable, even for relatively high-precision families, with very few descendants representing each ancestor. These observations suggest that revisiting these important protein domains using ASPEN's approach, including quantifying the likely accuracy of published trees and constructing and ranking trees most consistent with the available homolog sequences, is worthwhile. Specifically, we propose that for most protein families, it will be necessary to consider multiple equally likely models of evolutionary divergence.

## Materials and methods

### Sequences
#### Simulated paralog families
We simulated 600 families, each containing 15 paralogs, with each paralog represented by 66 orthologs. First, we generated random 15-leaf phylogenies representing paralog divergence. Random phylogenies were generated with average branch lengths of 0.5, 0.6, 0.7, 0.8, 0.9, and 1.0 –100 phylogenies each. Next, the Ensembl Compara species tree topology (*Herrero et al., 2016*) containing 66 metazoan species was grafted to each leaf of each random topology to represent ortholog divergence. Finally, each species tree topology was parametrized with branch lengths corresponding to species divergence times obtained from timetree.org (*Hedges et al., 2015*; *Kumar et al., 2017*), randomly rescaled in total height to represent faster or slower evolution of individual paralogs, and then had each individual segment randomly perturbed around its previous length. Sequence evolution was simulated over each resulting phylogeny, seeded with an alignment of human tyrosine kinase domains with median length of 269 a.a. All sequence simulation materials and simulated sequence alignments are available via Figshare (10.6084/m9.figshare.5263885).

#### LacI transcription factor family
We started with a collection of 19 LacI paralogs represented by 28 to 192 orthologs (*Sloutsky and Naegle, 2016*). After initial phylogenetic reconstruction we split paralogs PurR and RbsR-A into three separate paralogs each, according to monophyletic grouping of orthologs. This resulted in new paralogs PurR1 (37 orthologs), PurR2 (61 orthologs), PurR3 (28 orthologs), RbsR-A1 (79 orthologs), RbsR-A2 (22 orthologs), and RbsR-A3 (45 orthologs). The final collection contains 23 LacI paralogs represented by 22 to 192 orthologs, for a total of 1777 sequences (Supplementary Material).

### ASPEN topology reconstruction algorithm
#### Equivalence of topology representations
We demonstrate equivalence of acyclic graph and path length matrix representations of individual topologies by presenting a procedure for interconverting between the two. Transformation of a topology into its path lengths matrix representation is trivially accomplished by counting internal nodes along each path between pairs of leaves (*Figure 3A*). The reverse transformation can be accomplished using a simple bottom-up construction procedure. *Figure 3C* provides an illustration by reconstructing the topology from *Figure 3A* starting with its matrix representation. Because all path lengths are derived from a single topology, they are guaranteed to be consistent, making the construction unambiguous.

#### A branch-and-bound topology construction algorithm
Using the bottom-up procedure for reconstructing a single topology graph as a template, we developed an algorithm that uses a branch-and-bound strategy to construct the requested number of highest-scoring topologies according to the log-frequency scoring function (*Equation 1*). By analogy with the single-topology procedure, path lengths, together with their observation frequencies, are sorted into a list (*Figure 3D*). This list guides topology reconstruction (*Figure 3C*). However, unlike the single-topology case, list entries are not necessarily consistent with each other. The simplest illustration of this are paths of different lengths between the same two leaves, *Figure 3D*: for example $(A \leftrightarrow B, 1)$ observed in the ensemble 80% of the time and $(A \leftrightarrow B, 2)$ observed 20% of

the time. Such paths are clearly mutually exclusive. The branching component of branch-and-bound accommodates the divergent topologies which recapitulate each path.

## Branching

As in the single-topology procedure, construction of internal nodes is triggered by path length entries encountered during list traversal, with one key difference. In single topology reconstruction, if a path length could be recapitulated by the introduction of an internal node, that node could be safely constructed because it was guaranteed to satisfy every other list entry. Since that guarantee no longer holds, multiple topologies are constructed simultaneously by allowing the construction sequence to branch (*Figure 3—figure supplement 1*). "Assemblies'' are used to track simultaneous reconstruction of multiple topologies. Each assembly holds a copy of the path length frequencies list, a partially constructed topology, and the current topology score according to the scoring function. Reconstruction proceeds in iterations, starting with a single empty assembly on the first iteration. The entire list is traversed and every possible extension is created simultaneously in a copy of the original assembly. In each resulting assembly, all path lengths completed by the new node and all path lengths incompatible with it are marked and not re-examined on subsequent iterations. Remaining path lengths are not completed by the new node, but remain compatible with it. On subsequent iterations the same procedure is repeated for all tracked assemblies.

In principle, branching and iteration alone yield every topology consistent with path lengths observed in the ensemble. In practice, this results in a combinatorial explosion of tracked assemblies, which must be carefully managed to allow construction to proceed to completion.

First, branching to satisfy non-conflicting path lengths can lead to collisions between diverged construction sequences on later iterations (*Figure 3—figure supplement 1*). This occurs because most topologies can be constructed by introducing internal nodes in multiple orders. Each branched construction sequence represents a particular order of internal node introduction. In a practical implementation, these collisions must be managed in order to prevent construction of the same topology by multiple construction sequences – an enormous replication of effort.

Second, even if each distinct topology is constructed once, in most cases reconstructing every topology consistent with observations from the ensemble, no matter how infrequent, is neither practical nor useful. Bounding, described in the next section, guarantees reconstruction of only the requested number of top-scoring topologies.

## Bounding

Completed topologies are ranked according to their log-frequency score, with the ranking updated every time a new topology is finalized. The number of top scoring topologies to reconstruct, $N$, is specified at the beginning of a reconstruction run. Once the initial $N$ topologies have been constructed, the $N$th topology score constitutes the bound. Partially constructed topologies (assemblies) are abandoned if no complete topology can be derived from their construction state with a score above the current bound. We determine this by calculating the score for already-incorporated path lengths and projecting the best possible score for a complete topology by assuming the most frequent remaining path length will be incorporated for every unconnected leaf pair:

$$\text{projected score} = \sum_{incoprated\ paths} \log(f_{path^L}) + \sum_{remaining\ paths} \max_L(\log(f_{path^L})) \qquad (2)$$

As more high-scoring topologies are constructed, the bounding criterion becomes more strict, allowing both more and earlier abandoning of unproductive construction sequences. The branch-and-bound strategy guarantees that the $N$ topologies remaining on the list after all active assemblies have been completed or abandoned are the $N$ highest scoring topologies according to the scoring function.

## Simulation of sequence evolution

Random 15-leaf phylogenies were generated at www.trex.uqam.ca (*Boc et al., 2012*) using the procedure of *Kuhner and Felsenstein (1994)*. Human tyrosine kinase domains were aligned using MAFFT L-INS-i with default parameters. This alignment was used as the template for sequence

simulations as follows. The alignment was divided into 24 segments on the basis of local sequence similarity and analysis of solved tyrosine kinase structures. Each segment was assigned a substitution rate scaling factor and an insertion/deletion model to match degree of conservation and solvent exposure in solved structures. Simulation was carried out over synthetic phylogenies using indel-seq-gen (*Strope et al., 2007*; *Strope et al., 2009*; *Rambaut and Grass, 1997*) under the JTT substitution model.

All sequence simulation materials, including synthetic phylogenies, the template alignment, and indel-seq-gen control files, as well as simulated sequence alignments are available via Figshare (https://doi.org/10.6084/m9.figshare.5263885.v1).

## Phylogeny inference

All-sequence phylogenies were inferred using all combinations of MAFFT L-INS-i, ClustalOmega, and Muscle for sequence alignment with FastTree2 and RAxML for phylogeny inference. Sub-sampled phylogenies for precision calculations were inferred with FastTree2 only, due to run time considerations. Subsampled phylogenies for ensembles used by ASPEN were reconstructed using L-INS-i and FastTree2 only.

Alignment algorithms were used with their default settings. FastTree2 was used with default settings and the WAG substitution model. RAxML was used with default settings and the PROTGAM-MAWAGF variant of the WAG substitution model. The WAG substitution model was deliberately used for topology inference, instead of the JTT substitution model used for simulating protein families, in order to emulate the more realistic scenario where models used for reconstruction of phylogenies for natural families do not precisely match the substitution patterns in those families.

Accuracy and precision of reconstruction for a protein family are defined in terms of the L-INS-i/FastTree2 all-sequence and subsampled topologies.

## Assessment of alignment robustness

Robustness of sequence alignment to perturbations of alignment algorithm parameters (guide tree, sequence direction, and gap insertion penalty) was assessed with Guidance2 software (*Landan and Graur, 2007*; *Sela et al., 2015*). Because MAFFT L-INS-i was used for phylogeny inference, we assessed the robustness of alignment with this accuracy-oriented protocol, rather than the Guidance2 default, the speed-oriented MAFFT FFT-NS protocol. Because L-INS-i is substantially slower than FFT-NS, for each all-sequence alignment 15 alternative guide trees were used (resulting in 60 total alternative alignments) instead of the Guidance2 default of 100 alternative guide trees (400 total alternative alignments).

## Modified Robinson-Foulds topology distance metric

The Robinson-Foulds (*Robinson and Foulds, 1981*) ($RF$) metric is defined in terms of leaf partitions at internal topology nodes for two topologies with identical sets of leaves. For a tree with $N$ leaves there are $N - 3$ informative splits. The normalized form of the Robinson-Foulds comparison metric for two topologies, $A$ and $B$, is:

$$RF = \frac{x + y}{2N - 6} \tag{3}$$

where $x$ is the number of leaf partitions in $A$ but not in $B$, $y$ is the number of leaf partitions in $B$ but not in $A$, $N$ is the number of leaves in each topology, and $2N - 6 = 2 \times (N - 3)$ is the total number of informative splits in the two topologies.

In order to compare reconstructed ancestor divergence topologies, we had to modify the $RF$ metric to accommodate cases when the ancestor of a descendant sequence set has as descendants one or more other ancestors (non-monophyletic reconstruction). Such topologies are poorly formed because they require inference of additional unobservable events – loss of paralogs in some lineages – in order to be reconciled with a duplication/speciation divergence history. Because the offending subsample cannot be pruned to a common ancestor leaf, the resulting topology cannot be compared to properly formed topologies (e.g. the true topology) using the standard $RF$ metric. In effect, when sequence leaves and speciation internal nodes of the offending descendant set are pruned, the resulting topology is missing a leaf, because the corresponding ancestor maps to an internal

node. That node is ambiguous in its duplication vs speciation status. Nevertheless, internal nodes representing pre-duplication ancestors of the offending ancestor (and the descendant set representing it) and other ancestors of designated descendant sets can match equivalent nodes in other topologies in terms of induced partition of designated ancestors. Our modified version, $RF^{\star}$, can account for this.

In $RF^{\star}$, $N$ represents the number of designated ancestors (descendant sets) in each compared topology, not the number of leaves. In addition to $x$ and $y$ we define $z$ as the number of common ancestor leaves missing from $A$ but not from $B$ and $z'$ as the number of common ancestor leaves missing from $B$ but not from $A$. The modified metric is calculated as:

$$RF^{\star} = \frac{x+y+z+z'}{2N-6} \tag{4}$$

## ASPEN

ASPEN is implemented in python 2.7. The ASPEN development repository is publicly available at https://github.com/NaegleLab/ASPEN (*Sloutsky and Naegle, 2019*; copy archived at https://github.com/elifesciences-publications/ASPEN).

## Acknowledgements

This work was partially enabled by Center for Biological Systems Engineering. Computations were performed using the facilities of the Washington University Center for High Performance Computing, which were partially funded by NIH grants 1S10RR022984-01A1 and 1S10OD018091-01. Additionally, we would like to acknowledge Research Computing at The University of Virginia for providing computational resources and technical support that have contributed to the results reported within this publication. We wish to thank Dr. Tom Ronan, Dr. Barak Cohen, Dr. Gary Stormo, Dr. Justin Fay, and Dr. Jim Havranek for the helpful discussions that shaped this work, as well as editor Antonis Rokas and reviewers, Dr. Rama Ranganathan, Dr. Itamar Sela, Dr. Luke Wheeler, for their helpful suggestions.

## Additional information

### Funding

The funders had no role in study design, data collection and interpretation, or the decision to submit the work for publication.

### Author contributions

Roman Sloutsky, Conceptualization, Data curation, Software, Formal analysis, Validation, Investigation, Visualization, Methodology, Writing—original draft, Writing—review and editing; Kristen M Naegle, Conceptualization, Resources, Formal analysis, Supervision, Funding acquisition, Visualization, Methodology, Writing—original draft, Project administration, Writing—review and editing

### Author ORCIDs

Roman Sloutsky (iD) https://orcid.org/0000-0002-0794-1255
Kristen M Naegle (iD) https://orcid.org/0000-0001-7146-9592

### Decision letter and Author response
Decision letter https://doi.org/10.7554/eLife.47676.026
Author response https://doi.org/10.7554/eLife.47676.027

## Additional files

### Supplementary files
• Supplementary file 1. 1777 sequences from the LacI family, split among 23 orthosets, and the top 500 topologies reconstructed by ASPEN for the LacI family.

DOI: https://doi.org/10.7554/eLife.47676.021

• Transparent reporting form
DOI: https://doi.org/10.7554/eLife.47676.022

## Data availability

All input sequences are provided for simulated (http://doi.org/10.6084/m9.figshare.5263885) and LacI (supplementary material) protein families. Code used to perform ASPEN topology reconstruction is available at https://github.com/NaegleLab/ASPEN (copy archived at https://github.com/elifes-ciences-publications/ASPEN).

The following dataset was generated:

| Author(s) | Year | Dataset title | Dataset URL | Database and Identifier |
|---|---|---|---|---|
| Roman Sloutsky, Kristen Naegle | 2019 | Simulation Materials for ASPEN | http://doi.org/10.6084/m9.figshare.5263885 | figshare, 10.6084/m9.figshare.5263885 |

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
