## [Decision Letter]

Thank you for submitting your article "ASPEN: A methodology for reconstructing protein evolution with improved accuracy using ensemble models" for consideration by *eLife*. Your article has been reviewed by peer reviewers, and the evaluation has been overseen by a Reviewing Editor and Patricia Wittkopp as the Senior Editor. The following individuals involved in review of your submission have agreed to reveal their identity: Rama Ranganathan (Reviewer #1); Itamar Sela (Reviewer #2); Luke Wheeler (Reviewer #3).

The reviewers have discussed the reviews with one another and the Reviewing Editor has drafted this decision to help you prepare a revised submission.

Summary:

The study by Sloutsky and Neagle addresses the important problem of how to make good models of the evolutionary history of a protein family given extant sequences. This is a topical, important, and interesting problem, especially given the growing use of sequence databases for understanding the function and evolution of proteins. Protein sequences diverge through two processes that must be distinguished in making an accurate model of their history: speciation events which lead to orthologous sequences and duplication/divergence events that lead to paralogous sequences. The manuscript presents a meta-algorithm, ASPEN, that improves phylogenetic inference of protein families by subsampling sequences and constructing a topology which is supported by subsampled trees. Sampling alignment columns is a common practice which is used to assign bootstrap support values to phylogenetic trees splits, however, in this work the sampling is performed over rows (i.e. sequences). Notably, there is a correlation between "accuracy" (inferred topology quality) and "precision" (statistical support of inferred topology), as defined in the manuscript. Subsampled topologies are incorporated into a subtle branch-to-bound procedure, to construct topology that is statistically supported by many subsamples, rather than a single all-sequence alignment. The resulting constructed topology is shown to be more accurate than all-sequence based topology using a simulated dataset. The developed meta-algorithm will be of considerable interest for scientists that study evolution of protein/gene families and represents an advance in the effort to improve phylogenetic reconstruction by accounting for uncertainties in inference algorithms.

Essential revisions:

1) A key general point here is the real difficulty of actually inferring the true specific sequence of mutational events that underlie branches in an evolutionary tree. A very important and insightful concept that is partially expressed in this paper is to question the relevance of wanting to know that information from a point of view of deducing any principles of molecular evolution. The idea that the most invariant aspects of the tree reconstruction are simultaneously the most accurate is interesting and sensible, but isn't what is being proposed here that they are also the most relevant? If so, then this should be discussed and defended in more detail in the conclusions. In this same theme, what do these results mean for the whole community using ancestral reconstruction to infer epistasis along branches of the inferred trees and using such studies to extract principles of protein evolution? And for the interesting claims to having "resurrected" true ancestral states from all-sequence tree reconstructions? More generally, some discussion of what can be claimed and what cannot in reconstruction-based studies given the advances presented here about accuracy and precision in the process of these reconstructions would be valuable to guide the scientific community.

2) Analyses summarized in Figure 2 (and is also related to Figure 5). Figure 2A suggests that low accuracy topologies result from poor alignments, and are not due to failure of the tree reconstruction algorithm to retrieve the optimal topology for the given alignment. This can be tested directly by analysis of alignments quality. Moreover, since the quality of the inferred tree depends to a large extent on the quality of the alignment, such an analysis can provide a simple explanation for the observed correlation between accuracy and precision. We suggest that you compare the agreement of all-sequence and true alignments in several accuracy bins, as well as perform comparisons of subsampled alignments between themselves and to all-sequence and true alignments.

A plausible explanation for the accuracy-precision correlation is that different accuracy bins represent different levels of alignment problems complexities. Low complexity alignments are robust with respect to subsampling and are therefore associated with high precision. Complex alignment problems might produce very different subsampled alignments that result in different inferred topologies and low precision. Analysis of the alignments quality can directly support or disprove this hypothesis and better isolate the source of low accuracy (poor alignment or poor topology inference given the alignment).

Questions about how different scenarios might affect the efficacy of the ASPEN approach.

3) As noted in the paper, the simulated trees have a fixed number of paralogs. There are no new gene duplications that occur from the oldest ancestral state to the end-state used for reconstruction. How will variability in this feature in real biological trees effect the ASPEN reconstructions? What is the minimum number of "high confidence nodes" that one needs to make this work?

4) We wonder about using JTT model (or another similar amino acid substitution matrix) to conduct the simulated evolution, because it assumes independent evolution of each site/column. Real proteins will have epistatic interactions due to physical and functional constraints. Does this added complexity change the relationship between precision and accuracy or have some other effect on the efficacy of ASPEN? For example, will there be a deviation in the precision vs. accuracy curve for real proteins (like LacI in this study) relative to simulated proteins, that wouldn't be detectable without knowing the true model?

Question about software.

5) We were pleased with the availability of data and materials used in the paper via online repositories. One thing that would be very beneficial, though not necessary for publication of this article, is a thorough documentation of the ASPEN software that the authors have implemented. It will be difficult for any would-be users to dissect how the software functions without documentation. What are the authors' plans regarding providing documentation for the software?

---

## [Author Response]

Essential revisions:1) A key general point here is the real difficulty of actually inferring the true specific sequence of mutational events that underlie branches in an evolutionary tree. A very important and insightful concept that is partially expressed in this paper is to question the relevance of wanting to know that information from a point of view of deducing any principles of molecular evolution. The idea that the most invariant aspects of the tree reconstruction are simultaneously the most accurate is interesting and sensible, but isn't what is being proposed here that they are also the most relevant? If so, then this should be discussed and defended in more detail in the conclusions. In this same theme, what do these results mean for the whole community using ancestral reconstruction to infer epistasis along branches of the inferred trees and using such studies to extract principles of protein evolution? And for the interesting claims to having "resurrected" true ancestral states from all-sequence tree reconstructions? More generally, some discussion of what can be claimed and what cannot in reconstruction-based studies given the advances presented here about accuracy and precision in the process of these reconstructions would be valuable to guide the scientific community.

This is a wonderful summary of the problems phylogenetic inference presents. In fact, we appreciate this summary so much, we almost directly quoted the first line. We have added a new Conclusions section that summarizes this work in context of the field at large.

We have moved one of the original Discussion paragraphs to the new Conclusions section (covering example real protein family inferences that are ubiquitously used in the field and are likely inaccurate). Additionally, we have added the following two paragraphs to address the essence of the request here.

“Phylogenetic inference is an extremely difficult problem, particularly for individual protein families, where short amino acid sequences contain limited phylogenetic signal. Here we demonstrated that the sensitivity of inference and alignment algorithms to their inputs is a result of the failure of objective functions to accurately model the mutational events that underlie the divergence nodes in an evolutionary tree, which manifest in both alignment and inference procedures. At their core, phylogenetic trees are hypothetical models of a currently unknowable evolutionary history. George E.P. Box's much quoted philosophy about models is relevant here:

“Essentially, all models are wrong, but some are useful''. Extending this, we propose that integrating across a lot of models that are wrong in different ways, but right in the same way, yields more useful models of evolutionary divergence. This is based on the observation that topologies capturing more of the shared information across many incorrect models are significantly more accurate. However, we also find that, for especially difficult inference problems (where all-sequence topologies are imprecise, i.e. have low reproducibility under subsampling) it is important to consider multiple models as equally viable explanations of evolutionary history. In other words, for most real evolutionary inference problems, scientists should integrate their analyses over multiple plausible phylogenetic models. The methodology we presented facilitates both an overall assessment of accuracy of topology reconstruction for any particular problem, and, in light of that assessment, reconstruction of an appropriate number of maximally plausible topological models. Crucially, ASPEN is geared toward analysis of precisely the most difficult class ancestors: ancient ancestors of multiple paralogs.

It should be noted that individual ancestral nodes can vary dramatically in their difficulty of inference, and many ancestors of great biological interest can be inferred with very high accuracy, despite low overall accuracy of topology inference for that family. Moreover, given high confidence in the existence of such an ancestor, reconstruction of that ancestor's sequence is robust with respect to topological uncertainty for the clade below it [Eick et al., 2017]. A critical step in evolutionary analysis, then, is to determine *which* inferred ancestors are reliable – both with respect to systematic biases and with respect to uncertainty due to various sources of random noise – and which are not. ASPEN topologies can be used for this purpose in a fashion similar to the traditional phylogenetic bootstrap, with the additional benefit of evaluating consistency of an ancestor's inference across the most likely topologies.”

2) Analyses summarized in Figure 2 (and is also related to Figure 5). Figure 2A suggests that low accuracy topologies result from poor alignments, and are not due to failure of the tree reconstruction algorithm to retrieve the optimal topology for the given alignment. This can be tested directly by analysis of alignments quality. Moreover, since the quality of the inferred tree depends to a large extent on the quality of the alignment, such an analysis can provide a simple explanation for the observed correlation between accuracy and precision. We suggest that you compare the agreement of all-sequence and true alignments in several accuracy bins, as well as perform comparisons of subsampled alignments between themselves and to all-sequence and true alignments.A plausible explanation for the accuracy-precision correlation is that different accuracy bins represent different levels of alignment problems complexities. Low complexity alignments are robust with respect to subsampling and are therefore associated with high precision. Complex alignment problems might produce very different subsampled alignments that result in different inferred topologies and low precision. Analysis of the alignments quality can directly support or disprove this hypothesis and better isolate the source of low accuracy (poor alignment or poor topology inference given the alignment).

Thanks for the great suggestions. At the level of perturbation we are performing, both alignment and topology search are affected and interconnected in their ability to identify models that best describe the data (be it alignment or a tree based on that alignment). In the first draft of the manuscript we knew that alignment error alone cannot explain the phenomenon identified here, since, for families with higher precision, topology inference cannot identify the true topology, given the true alignment: the all-sequence topology (inferred from the all-sequence L-INS-i alignment under the WAG model) has higher likelihood under the JTT model and the true alignment than does the true topology. Under these conditions the scoring procedure in the topology inference algorithm scores inaccurate topologies as better than true topologies, even when given the true alignment and substitution model. We did not sufficiently describe those results in the original manuscript and have expanded on that point.

We prefer the term “alignment robustness” to “alignment quality”, because the latter seems too subjective. After all, different alignments can serve better or worse for different purposes. In order to address the root question here – can alignment robustness explain the observed relationship between topology precision and topology accuracy? – we performed the suggested experiment and compared the all-seq alignments to the true alignments (we call this alignment accuracy) using the Guidance/Guidance2 sum-of-pairs similarity metric. Indeed, there is a correlation between alignment accuracy and topology accuracy (Pearson Correlation of 0.80), but this correlation is weaker than that of topology precision and topology accuracy (Pearson Correlation of 0.91). The lower correlation is driven by the large deviations in alignment accuracy and a compression in this scale – no all-sequence alignment is completely accurate, yet these can ultimately yield completely accurate trees, and most of the mid-accuracy/mid-precision trees come from all-sequence alignments that are almost wholly different than the true alignment.

Comparing alignments of subsampled sequences to each other or to the all-sequence or true alignment is problematic, as each subsample has a different collection of sequences. We were concerned that resulting similarity might conflate the intended understanding with the additional variability of randomness of the characteristics in that particular subsample. Instead, we asked whether the reproducibility of the all-sequence alignment (we call this alignment robustness) could explain reproducibility of inferred topologies (precision). We used Guidance2 to measure the all-sequence alignment robustness and found that this is almost perfectly correlated with the all-sequence alignment accuracy (Pearson’s Correlation of 0.97), but does not capture any additional variance of topology accuracy (Pearson’s correlation between alignment precision and tree accuracy is 0.78). Hence, one could use the reproducibility of the all-sequence alignment to predict the accuracy of the all-sequence alignment (a hidden variable) to a great extent, but that value does not translate as precisely to the resulting topology accuracy as does topology precision. We believe this is likely due to the nuances of how and where alignment errors are located and how they ultimately influence the resulting topology.

In short, the answer to the question asked by the reviewers is yes – alignment accuracy correlates with tree accuracy and, since alignment reproducibility correlates almost perfectly with alignment accuracy, alignment reproducibility also correlates with topology accuracy. However, an emergent relationship exists between topology precision and topology accuracy that gives additional predictive power to topology precision that is not fully explained by either alignment accuracy or alignment reproducibility.

We have added a new figure and edited the section “Subsampling produces an observable measure of accuracy” to clarify the roles and contributions of alignment vs. topology scoring in the precision/accuracy.

Questions about how different scenarios might affect the efficacy of the ASPEN approach.3) As noted in the paper, the simulated trees have a fixed number of paralogs. There are no new gene duplications that occur from the oldest ancestral state to the end-state used for reconstruction. How will variability in this feature in real biological trees effect the ASPEN reconstructions? What is the minimum number of "high confidence nodes" that one needs to make this work?

It is certainly true that real protein families may have additional complexity of speciations, duplications, and gene loss that are not reflected in the specific set of evolutionary circumstances under which we simulated evolution. However, these nuances of duplication/speciation/loss will not affect ASPEN reconstructions, unless one has consistently mis-assigned or cannot confidently assign such a paralog to a node – i.e. it only matters whether subsampling under that node produces consistently monophyletic topology reconstructions. The nature of the node does affect the analysis or the reconstruction.

The Results subsection “Defining protein families for testing” is expanded, and now states:

“Evolution of real protein families is rarely that neat. Speciations can predate duplications. Loss of some paralogs in some lineages can result in different collections of paralogs appearing in different genomes, dramatically complicating orthology assignments. […] Therefore, the analysis based on this simulation experiment is applicable without loss of generality.”

Additionally, we have expanded theDiscussion subsection called “Selection of ancestral nodes for topology search space decomposition”,detailing of the impacts and implications of selection of nodes. We also provide some practical recommendations for identifying high-confidence nodes and handling poorly-behaved leaf sequences.

4) We wonder about using JTT model (or another similar amino acid substitution matrix) to conduct the simulated evolution, because it assumes independent evolution of each site/column. Real proteins will have epistatic interactions due to physical and functional constraints. Does this added complexity change the relationship between precision and accuracy or have some other effect on the efficacy of ASPEN? For example, will there be a deviation in the precision vs. accuracy curve for real proteins (like LacI in this study) relative to simulated proteins, that wouldn't be detectable without knowing the true model?

This is a really interesting and complex question. We believe that a detailed answer to this question requires a careful and extensive study beyond the scope of this manuscript. Answering this question requires using a generative model that incorporates constraints amongst sequence sites reflective of constraints experienced by real protein families. To date, all attempts at this have relied on physical models of protein structure with rather simplistic assumptions (e.g. that the only constraints arise from protein folding requirements) [Arenas et al., 2013; Nasrallah et al., 2011; Grahnen et al., 2011; Choi et al., 2007; Rodrigue et al., 2005; Robinson et al., 2003; Parisi et al., 2001]. Despite their reductionism, are far too computationally intensive for the kind of large-scale sequence evolution simulation we carried out and for phylogenetic reconstruction of the simulated families. In addition, at the moment these models are not available as implementations that can be incorporated into the ASPEN analysis workflow.

The study of protein evolution as a whole would tremendously benefit from a more realistic treatment of evolutionary constraints experienced by proteins than what is offered by the commonly used substitution models. Algorithmic improvements to make physics based-models more tractable and incorporation of such models into popular sequence simulation and phylogenetic inference software will be necessary to make them more penetrant in the field.

At this point, we have included our thoughts on this process as part of the new Discussion subsection “Generality of the accuracy-precision relationship”, where we propose that non-independence would ultimately manifest as less overall information on which to build the model, leading to a more complex/difficult inference problem, but not substantively affecting the accuracy-precision relationship.

New Discussion subsection titled “Generality of the accuracy-precision relationship” covers this topic and others.

Question about software.5) We were pleased with the availability of data and materials used in the paper via online repositories. One thing that would be very beneficial, though not necessary for publication of this article, is a thorough documentation of the ASPEN software that the authors have implemented. It will be difficult for any would-be users to dissect how the software functions without documentation. What are the authors' plans regarding providing documentation for the software?

Thank you for the suggestion. We have created a clean repository to accompany this publication (moving the development code to another repository in GitHub) and are in the process of documenting these tools. We did not wish to delay the evaluation of this revision and so this new repository is not yet fully populated. However, it will be complete by the end of September. In this clean repository, we will focus on the most complex aspect of this project: converting many inferred trees into a pairwise distance distribution and performing ASPEN reconstruction of best topologies according that distribution. We will also provide some scripts as examples of how to subsample for the creation of the family of trees. However, we highly recommend users to select their own methods for alignment and inference, and subset size as they see fit. In addition, we hope that upon publication of this manuscript, our ability to secure future funding for development of ASPEN will be improved, allowing us to improve usability and flexibility of ASPEN as a software tool.